# GDDIM: GENERALIZED DENOISING DIFFUSION IMPLICIT MODELS

**Qinsheng Zhang**
Georgia Institute of Technology
qzhang419@gatech.edu

**Molei Tao**
Georgia Institute of Technology
mtao@gatech.edu

**Yongxin Chen**
Georgia Institute of Technology
yongchen@gatech.edu

## ABSTRACT

Our goal is to extend the denoising diffusion implicit model (DDIM) to general diffusion models (DMs) besides isotropic diffusions. Instead of constructing a non-Markov noising process as in the original DDIM, we examine the mechanism of DDIM from a numerical perspective. We discover that the DDIM can be obtained by using some specific approximations of the score when solving the corresponding stochastic differential equation. We present an interpretation of the accelerating effects of DDIM that also explains the advantages of a deterministic sampling scheme over the stochastic one for fast sampling. Building on this insight, we extend DDIM to general DMs, coined generalized DDIM (gDDIM), with a small but delicate modification in parameterizing the score network. We validate gDDIM in two non-isotropic DMs: Blurring diffusion model (BDM) and Critically-damped Langevin diffusion model (CLD). We observe more than 20 times acceleration in BDM. In the CLD, a diffusion model by augmenting the diffusion process with velocity, our algorithm achieves an FID score of 2.26, on CIFAR10, with only 50 number of score function evaluations (NFEs) and an FID score of 2.86 with only 27 NFEs. Project page and code: https://github.com/qsh-zh/gDDIM.

## 1 INTRODUCTION

Generative models based on diffusion models (DMs) have experienced rapid developments in the past few years and show competitive sample quality compared with generative adversarial networks (GANs) (Dhariwal & Nichol, 2021; Ramesh et al.; Rombach et al., 2021), competitive negative log likelihood compared with autoregressive models in various domains and tasks (Song et al., 2021; Kawar et al., 2021). Besides, DMs enjoy other merits such as stable and scalable training, and mode-collapsing resiliency (Song et al., 2021; Nichol & Dhariwal, 2021). However, slow and expensive sampling prevents DMs from further application in more complex and higher dimension tasks. Once trained, GANs only forward pass neural networks once to generate samples, but the vanilla sampling method of DMs needs 1000 or even 4000 steps (Nichol & Dhariwal, 2021; Ho et al., 2020; Song et al., 2020b) to pull noise back to the data distribution, which means thousands of neural networks forward evaluations. Therefore, the generation process of DMs is several orders of magnitude slower than GANs.

How to speed up sampling of DMs has received significant attention. Building on the seminal work by Song et al. (2020b) on the connection between stochastic differential equations (SDEs) and diffusion models, a promising strategy based on *probability flows* (Song et al., 2020b) has been developed. The probability flows are ordinary differential equations (ODE) associated with DMs that share equivalent marginal with SDE. Simple plug-in of off-the-shelf ODE solvers can already achieve significant acceleration compared to SDEs-based methods (Song et al., 2020b). The arguably most popular sampling method is *denoising diffusion implicit model (DDIM)* (Song et al., 2020a), which includes both deterministic and stochastic samplers, and both show tremendous im-

provement in sampling quality compared with previous methods when only a small number of steps is used for the generation.

Although significant improvements of the DDIM in sampling efficiency have been observed empirically, the understanding of the mechanism of the DDIM is still lacking. First, why does solving probability flow ODE provide much higher sample quality than solving SDEs, when the number of steps is small? Second, it is shown that stochastic DDIM reduces to marginal-equivalent SDE (Zhang & Chen, 2022), but its discretization scheme and mechanism of acceleration are still unclear. Finally, can we generalize DDIMs to other DMs and achieve similar or even better acceleration results?

In this work, we conduct a comprehensive study to answer the above questions, so that we can generalize and improve DDIM. We start with an interesting observation that the DDIM can solve corresponding SDEs/ODE **exactly** without any discretization error in finite or even one step when the training dataset consists of only one data point. For deterministic DDIM, we find that the added noise in perturbed data along the diffusion is constant along an exact solution of probability flow ODE (see Prop 1). Besides, provided only one evaluation of log density gradient (a.k.a. score), we are already able to recover accurate score information for any datapoints, and this explains the acceleration of stochastic DDIM for SDEs (see Prop 3). Based on this observation, together with the manifold hypothesis, we present one possible interpretation to explain why the discretization scheme used in DDIMs is effective on realistic datasets (see Fig. 2). Equipped with this new interpretation, we extend DDIM to general DMs, which we coin *generalized DDIM (gDDIM)*. With only a small but delicate change of the score model parameterization during sampling, gDDIM can accelerate DMs based on general diffusion processes. Specifically, we verify the sampling quality of gDDIM on Blurring diffusion models (BDM) (Hoogeboom & Salimans, 2022; Rissanen et al., 2022) and critically-damped Langevin diffusion (CLD) (Dockhorn et al., 2021) in terms of Fréchet inception distance (FID) (Heusel et al., 2017).

To summarize, we have made the following contributions: 1) We provide an interpretation for the DDIM and unravel its mechanism. 2) The interpretation not only justifies the numerical discretization of DDIMs but also provides insights into why ODE-based samplers are preferred over SDE-based samplers when NFE is low. 3) We propose gDDIM, a generalized DDIM that can accelerate a large class of DMs deterministically and stochastically. 4) We show by extensive experiments that gDDIM can drastically improve sampling quality/efficiency almost for free. Specifically, when applied to CLD, gDDIM can achieve an FID score of 2.86 with only 27 steps and 2.26 with 50 steps. gDDIM has more than 20 times acceleration on BDM compared with the original samplers.

The rest of this paper is organized as follows. In Sec. 2 we provide a brief inntroduction to diffusion models. In Sec. 3 we present an interpretation of the DDIM that explains its effectiveness in practice. Built on this interpretation, we generalize DDIM for general diffusion models in Sec. 4.

## 2 BACKGROUND

In this section, we provide a brief introduction to diffusion models (DMs). Most DMs are built on two diffusion processes in continuous-time, one forward diffusion known as the noising process that drives any data distribution to a tractable distribution such as Gaussian by gradually adding noise to the data, and one backward diffusion known as the denoising process that sequentially removes noise from noised data to generate realistic samples. The continuous-time noising and denoising processes are modeled by stochastic differential equations (SDEs) (Särkkä & Solin, 2019).

In particular, the forward diffusion is a linear SDE with state $\boldsymbol{u}(t) \in \mathbb{R}^D$

$$d\boldsymbol{u} = \boldsymbol{F}_t\boldsymbol{u}dt + \boldsymbol{G}_t d\boldsymbol{w}, t \in [0, T] \tag{1}$$

where $\boldsymbol{F}_t, \boldsymbol{G}_t \in \mathbb{R}^{D \times D}$ represent the linear drift coefficient and diffusion coefficient respectively, and $\boldsymbol{w}$ is a standard Wiener process. When the coefficients are piece-wise continuous, Eq. (1) admits a unique solution (Oksendal, 2013). Denote by $p_t(\boldsymbol{u})$ the distribution of the solutions $\{\boldsymbol{u}(t)\}_{0 \leq t \leq T}$ (simulated trajectories) to Eq. (1) at time $t$, then $p_0$ is determined by the data distribution and $p_T$ is a (approximate) Gaussian distribution. That is, the forward diffusion Eq. (1) starts as a data sample and ends as a Gaussian random variable. This can be achieved with properly chosen coefficients $\boldsymbol{F}_t, \boldsymbol{G}_t$. Thanks to linearity of Eq. (1), the transition probability $p_{st}(\boldsymbol{u}(t)|\boldsymbol{u}(s))$ from $\boldsymbol{u}(s)$ to $\boldsymbol{u}(t)$ is a Gaussian distribution. For convenience, denote $p_{0t}(\boldsymbol{u}(t)|\boldsymbol{u}(0))$ by $\mathcal{N}(\mu_t\boldsymbol{u}(0), \Sigma_t)$ where $\mu_t, \Sigma_t \in \mathbb{R}^{D \times D}$.

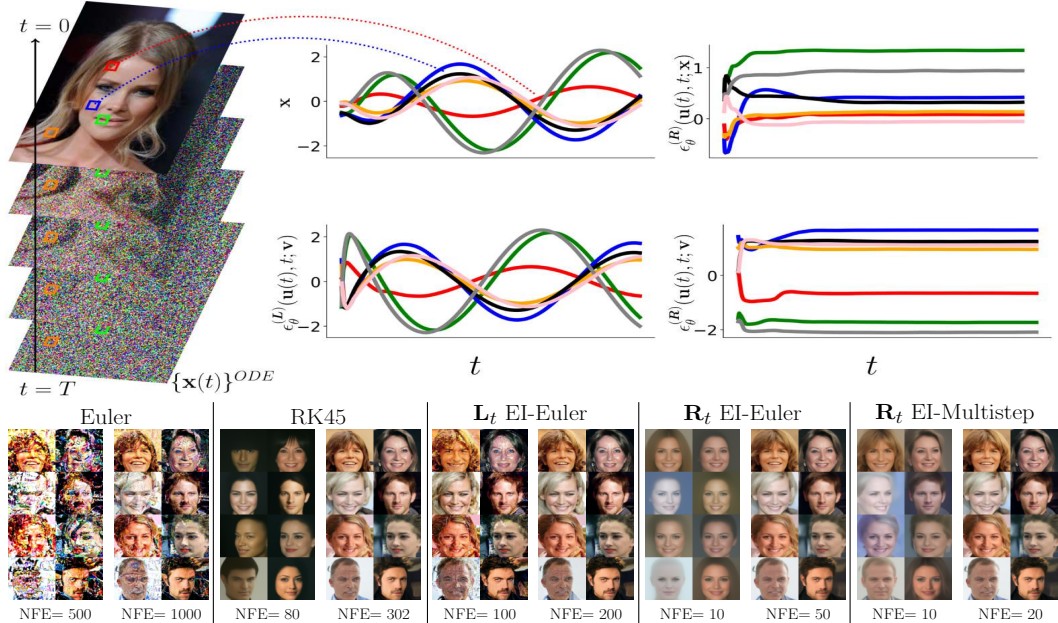

Figure 1: Importance of $\boldsymbol{K}_t$ for score parameterization $\boldsymbol{s}_\theta(\boldsymbol{u}, t) = -\boldsymbol{K}_t^{-T}\epsilon_\theta(\boldsymbol{u}, t)$ and acceleration of diffusion sampling with probability flow ODE. Trajectories of probability ODE for CLD (Dockhorn et al., 2021) at random pixel locations (Left). Pixel value and output of $\epsilon_\theta$ in $\boldsymbol{v}$ channel with choice $\boldsymbol{K}_t = \boldsymbol{L}_t$ (Dockhorn et al., 2021) along the trajectory (Mid). Output of $\epsilon_\theta$ in $\boldsymbol{x}, \boldsymbol{v}$ channels with our choice $\boldsymbol{R}_t$ (Right). The smooth network output along trajectories enables large stepsize and thus sampling acceleration. gDDIM based on the proper parameterization of $\boldsymbol{K}_t$ can accelerate more than 50 times compared with the naive Euler solver (Lower row).

The backward process from $\boldsymbol{u}(T)$ to $\boldsymbol{u}(0)$ of Eq. (1) is the denoising process. It can be characterized by the backward SDE simulated in reverse-time direction (Song et al., 2020b; Anderson, 1982)

$$d\boldsymbol{u} = [\boldsymbol{F}_t\boldsymbol{u}dt - \boldsymbol{G}_t\boldsymbol{G}_t^T\nabla\log p_t(\boldsymbol{u})]dt + \boldsymbol{G}_td\bar{\boldsymbol{w}}, \tag{2}$$

where $\bar{\boldsymbol{w}}$ denotes a standard Wiener process running backward in time. Here $\nabla\log p_t(\boldsymbol{u})$ is known as the score function. When Eq. (2) is initialized with $\boldsymbol{u}(T) \sim p_T$, the distribution of the simulated trajectories coincides with that of the forward diffusion Eq. (1). Thus, $\boldsymbol{u}(0)$ of these trajectories are unbiased samples from $p_0$; the backward diffusion Eq. (2) is an ideal generative model.

In general, the score function $\nabla\log p_t(\boldsymbol{u})$ is not accessible. In diffusion-based generative models, a time-dependent network $\boldsymbol{s}_\theta(\boldsymbol{u}, t)$, known as the score network, is used to fit the score $\nabla\log p_t(\boldsymbol{u})$. One effective approach to train $\boldsymbol{s}_\theta(\boldsymbol{u}, t)$ is the denoising score matching (DSM) technique (Song et al., 2020b; Ho et al., 2020; Vincent, 2011) that seeks to minimize the DSM loss

$$\mathbb{E}_{t\sim\mathcal{U}[0,T]}\mathbb{E}_{\boldsymbol{u}(0),\boldsymbol{u}(t)|\boldsymbol{u}(0)}[\|\nabla\log p_{0t}(\boldsymbol{u}(t)|\boldsymbol{u}(0)) - \boldsymbol{s}_\theta(\boldsymbol{u}(t), t)\|_{\Lambda_t}^2], \tag{3}$$

where $\mathcal{U}[0, T]$ represents the uniform distribution over the interval $[0, T]$. The time-dependent weight $\Lambda_t$ is chosen to balance the trade-off between sample fidelity and data likelihood of learned generative model (Song et al., 2021). It is discovered in Ho et al. (2020) that reparameterizing the score network by

$$\boldsymbol{s}_\theta(\boldsymbol{u}, t) = -\boldsymbol{K}_t^{-T}\epsilon_\theta(\boldsymbol{u}, t) \tag{4}$$

with $\boldsymbol{K}_t\boldsymbol{K}_t^T = \Sigma_t$ leads to better sampling quality. In this parameterization, the network tries to predict directly the noise added to perturb the original data. Invoking the expression $\mathcal{N}(\mu_t\boldsymbol{u}(0), \Sigma_t)$ of $p_{0t}(\boldsymbol{u}(t)|\boldsymbol{u}(0))$, this parameterization results in the new DSM loss

$$\mathcal{L}(\theta) = \mathbb{E}_{t\sim\mathcal{U}[0,T]}\mathbb{E}_{\boldsymbol{u}(0)\sim p_0,\epsilon\sim\mathcal{N}(0,\boldsymbol{I}_D)}[\|\epsilon - \epsilon_\theta(\mu_t\boldsymbol{u}(0) + \boldsymbol{K}_t\epsilon, t)\|_{\boldsymbol{K}_t^{-1}\Lambda_t\boldsymbol{K}_t^{-T}}^2]. \tag{5}$$

**Sampling:** After the score network $\boldsymbol{s}_\theta$ is trained, one can generate samples via the backward SDE Eq. (2) with a learned score, or the marginal equivalent SDE/ODE (Song et al., 2020b; Zhang &

Chen, 2021; 2022)

$$du = [\boldsymbol{F}_t \boldsymbol{u} - \frac{1+\lambda^2}{2} \boldsymbol{G}_t \boldsymbol{G}_t^T \boldsymbol{s}_\theta(\boldsymbol{u}, t)]dt + \lambda \boldsymbol{G}_t d\boldsymbol{w}, \tag{6}$$

where $\lambda \geq 0$ is a free parameter. Regardless of the value of $\lambda$, the exact solutions to Eq. (6) produce unbiased samples from $p_0(\boldsymbol{u})$ if $\boldsymbol{s}_\theta(\boldsymbol{u}, t) = \nabla \log p_t(\boldsymbol{u})$ for all $t, \boldsymbol{u}$. When $\lambda = 1$, Eq. (6) reduces to reverse-time diffusion in Eq. (2). When $\lambda = 0$, Eq. (6) is known as the *probability flow ODE* (Song et al., 2020b)

$$du = [\boldsymbol{F}_t \boldsymbol{u} - \frac{1}{2} \boldsymbol{G}_t \boldsymbol{G}_t^T \boldsymbol{s}_\theta(\boldsymbol{u}, t)]dt. \tag{7}$$

**Isotropic diffusion and DDIM:** Most existing DMs are isotropic diffusions. A popular DM is *Denoising diffusion probabilistic modeling (DDPM)* (Ho et al., 2020). For a given data distribution $p_{\text{data}}(\boldsymbol{x})$, DDPM has $\boldsymbol{u} = \boldsymbol{x} \in \mathbb{R}^d$ and sets $p_0(\boldsymbol{u}) = p_{\text{data}}(\boldsymbol{x})$. Though originally proposed in the discrete-time setting, it can be viewed as a discretization of a continuous-time SDE with parameters

$$\boldsymbol{F}_t := \frac{1}{2} \frac{d \log \alpha_t}{dt} \boldsymbol{I}_d, \quad \boldsymbol{G}_t := \sqrt{-\frac{d \log \alpha_t}{dt}} \boldsymbol{I}_d \tag{8}$$

for a decreasing scalar function $\alpha_t$ satisfying $\alpha_0 = 1, \alpha_T = 0$. Here $\boldsymbol{I}_d$ represents the identity matrix of dimension $d$. For this SDE, $\boldsymbol{K}_t$ is always chosen to be $\sqrt{1 - \alpha_t} \boldsymbol{I}_d$.

The sampling scheme proposed in DDPM is inefficient; it requires hundreds or even thousands of steps, and thus number of score function evaluations (NFEs), to generate realistic samples. A more efficient alternative is the *Denoising diffusion implicit modeling (DDIM)* proposed in Song et al. (2020a). It proposes a different sampling scheme over a grid $\{t_i\}$

$$\boldsymbol{x}(t_{i-1}) = \sqrt{\frac{\alpha_{t_{i-1}}}{\alpha_{t_i}}} \boldsymbol{x}(t_i) + (\sqrt{1 - \alpha_{t_{i-1}} - \sigma_{t_i}^2} - \sqrt{1 - \alpha_{t_i}} \sqrt{\frac{\alpha_{t_{i-1}}}{\alpha_{t_i}}}) \epsilon_\theta(\boldsymbol{x}(t_i), t_i) + \sigma_{t_i} \epsilon, \tag{9}$$

where $\{\sigma_{t_i}\}$ are hyperparameters and $\epsilon \sim \mathcal{N}(0, \boldsymbol{I}_d)$. DDIM can generate reasonable samples within 50 NFEs. For the special case where $\sigma_{t_i} = 0$, it is recently discovered in Zhang & Chen (2022) that Eq. (9) coincides with the numerical solution to Eq. (7) using an advanced discretization scheme known as the *exponential integrator (EI)* that utilizes the semi-linear structure of Eq. (7).

**CLD and BDM:** Dockhorn et al. (2021) propose *critically-damped Langevin diffusion (CLD)*, a DM based on an augmented diffusion with an auxiliary velocity term. More specifically, the state of the diffusion in CLD is of the form $\boldsymbol{u}(t) = [\boldsymbol{x}(t), \boldsymbol{v}(t)] \in \mathbb{R}^{2d}$ with velocity variable $\boldsymbol{v}(t) \in \mathbb{R}^d$. The CLD employs the forward diffusion Eq. (1) with coefficients

$$\boldsymbol{F}_t := \begin{bmatrix} 0 & \beta M^{-1} \\ \beta & -\Gamma\beta M^{-1} \end{bmatrix} \otimes \boldsymbol{I}_d, \quad \boldsymbol{G}_t := \begin{bmatrix} 0 & 0 \\ 0 & -\Gamma\beta M^{-1} \end{bmatrix} \otimes \boldsymbol{I}_d. \tag{10}$$

Here $\Gamma > 0, \beta > 0, M > 0$ are hyperparameters. Compared with most other DMs such as DDPM that inject noise to the data state $\boldsymbol{x}$ directly, the CLD introduces noise to the data state $\boldsymbol{x}$ through the coupling between $\boldsymbol{v}$ and $\boldsymbol{x}$ as the noise only affects the velocity component $\boldsymbol{v}$ directly. Another interesting DM is *Blurring diffusion model* (BDM) (Hoogeboom & Salimans, 2022). It can be shown the forward process in BDM can be formulated as a SDE with (Detailed derivation in App. B)

$$\boldsymbol{F}_t := \frac{d \log[\boldsymbol{V}\boldsymbol{\alpha}_t\boldsymbol{V}^T]}{dt}, \quad \boldsymbol{G}_t := \sqrt{\frac{d\boldsymbol{\sigma}_t^2}{dt} - \boldsymbol{F}_t\boldsymbol{\sigma}_t^2 - \boldsymbol{\sigma}_t^2\boldsymbol{F}_t}, \tag{11}$$

where $\boldsymbol{V}^T$ denotes a Discrete Cosine Transform (DCT) and $\boldsymbol{V}$ denotes the Inverse DCT. Diagonal matrices $\boldsymbol{\alpha}_t, \boldsymbol{\sigma}_t$ are determined by frequencies information and dissipation time. Though it is argued that inductive bias in CLD and BDM can benefit diffusion model (Dockhorn et al., 2021; Hoogeboom & Salimans, 2022), non-isotropic DMs are not easy to accelerate. Compared with DDPM, CLD introduces significant oscillation due to $\boldsymbol{x}$-$\boldsymbol{v}$ coupling while only inefficient ancestral sampling algorithm supports BDM (Hoogeboom & Salimans, 2022).

# 3 REVISIT DDIM: GAP BETWEEN THE EXACT SOLUTION AND NUMERICAL SOLUTION

The complexity of sampling from a DM is proportional to the NFEs used to numerically solve Eq. (6). To establish a sampling algorithm with a small NFEs, we ask the bold question:

*Can we generate samples **exactly** from a DM with finite steps if the score function is precise?*

To gain some insights into this question, we start with the simplest scenario where the training dataset consists of only one data point $x_0$. It turns out that accurate sampling from diffusion models on this toy example is not that easy, even if the exact score function is accessible. Most well-known numerical methods for Eq. (6), such as Runge Kutta (RK) for ODE, Euler-Maruyama (EM) for SDE, are accompanied by discretization error and cannot recover the single data point in the training set unless an infinite number of steps are used. Surprisingly, DDIMs can recover the single data point in this toy example in one step.

Built on this example, we show how the DDIM can be obtained by solving the SDE/ODE Eq. (6) with proper approximations. The effectiveness of DDIM is then explained by justifying the usage of those approximations for general datasets at the end of this section.

**ODE sampling**    We consider the deterministic DDIM, that is, Eq. (9) with $\sigma_{t_i} = 0$. In view of Eq. (8), the score network Eq. (4) is $s_\theta(u,t) = -\frac{\epsilon_\theta(u,t)}{\sqrt{1-\alpha_t}}$. To differentiate between the learned score and the real score, denote the ground truth version of $\epsilon_\theta$ by $\epsilon_{\mathrm{GT}}$. In our toy example, the following property holds for $\epsilon_{\mathrm{GT}}$.

**Proposition 1.** *Assume $p_0(u)$ is a Dirac distribution. Let $u(t)$ be an arbitrary solution to the probability flow ODE Eq. (7) with coefficient Eq. (8) and the ground truth score, then $\epsilon_{\mathrm{GT}}(u(t), t) = -\sqrt{1-\alpha_t} \nabla \log p_t(u(t))$ remains constant, which is $\nabla \log p_T(u(T))$, along $u(t)$.*

We remark that even though $\epsilon_{\mathrm{GT}}(u(t), t)$ remains constant along an exact solution, the score $\nabla \log p_t(u(t))$ is time-varying. This underscores the advantage of the parameterization $\epsilon_\theta$ over $s_\theta$. Inspired by Prop 1, we devise a sampling algorithm as follows that can recover the exact data point in one step for our toy example. This algorithm turns out to coincide with the deterministic DDIM.

**Proposition 2.** *With the parameterization $s_\theta(u, \tau) = -\frac{\epsilon_\theta(u,\tau)}{\sqrt{1-\alpha_\tau}}$ and the approximation $\epsilon_\theta(u, \tau) \approx \epsilon_\theta(u(t), t)$ for $\tau \in [t - \Delta t, t]$, the solution to the probability flow ODE Eq. (7) with coefficient Eq. (8) is*

$$u(t - \Delta t) = \sqrt{\frac{\alpha_{t-\Delta t}}{\alpha_t}} u(t) + \left( \sqrt{1 - \alpha_{t-\Delta t}} - \sqrt{1 - \alpha_t} \sqrt{\frac{\alpha_{t-\Delta t}}{\alpha_t}} \right) \epsilon_\theta(u(t), t), \qquad (12)$$

*which coincides with deterministic DDIM.*

When $\epsilon_\theta = \epsilon_{\mathrm{GT}}$ as is the case in our toy example, there is no approximation error in Prop 2 and Eq. (12) is precise. This implies that deterministic DDIM can recover the training data in one step in our example. The update Eq. (12) corresponds to a numerical method known as the exponential integrator to the probability flow ODE Eq. (7) with coefficient Eq. (8) and parameterization $s_\theta(u, \tau) = -\frac{\epsilon_\theta(u,\tau)}{\sqrt{1-\alpha_\tau}}$. This strategy is used and developed recently in Zhang & Chen (2022). Prop 1 and toy experiments in Fig. 2 provide sights on why such a strategy should work.

**SDE sampling**    The above discussions however do not hold for stochastic cases where $\lambda > 0$ in Eq. (6) and $\sigma_{t_i} > 0$ in Eq. (9). Since the solutions to Eq. (6) from $t = T$ to $t = 0$ are stochastic, neither $\nabla \log p_t(u(t))$ nor $\epsilon_{\mathrm{GT}}(u(t), t)$ remains constant along sampled trajectories; both are affected by the stochastic noise. The denoising SDE Eq. (6) is more challenging compared with the probability ODE since it injects additional noise to $u(t)$. The score information needs to remove not only noise presented in $u(T)$ but also injected noise along the diffusion. In general, one evaluation of $\epsilon_\theta(u, t)$ can only provide the information to remove noise in the current state $u$; it cannot predict the future injected noise. Can we do better? The answer is affirmative on our toy dataset. Given only one score evaluation, it turns out that score at any point can be recovered.

**Proposition 3.** *Assume SDE coefficients Eq. (8) and that $p_0(u)$ is a Dirac distribution. Given any evaluation of the score function $\nabla \log p_s(u(s))$, one can recover $\nabla \log p_t(u)$ for any $t, u$ as*

$$\nabla \log p_t(u) = \frac{1 - \alpha_s}{1 - \alpha_t} \sqrt{\frac{\alpha_t}{\alpha_s}} \nabla \log p_s(u(s)) - \frac{1}{1 - \alpha_t} \left( u - \sqrt{\frac{\alpha_t}{\alpha_s}} u(s) \right). \qquad (13)$$

The major difference between Prop 3 and Prop 1 is that Eq. (13) retains the dependence of the score over the state $u$. This dependence is important in canceling the injected noise in the denoising SDE

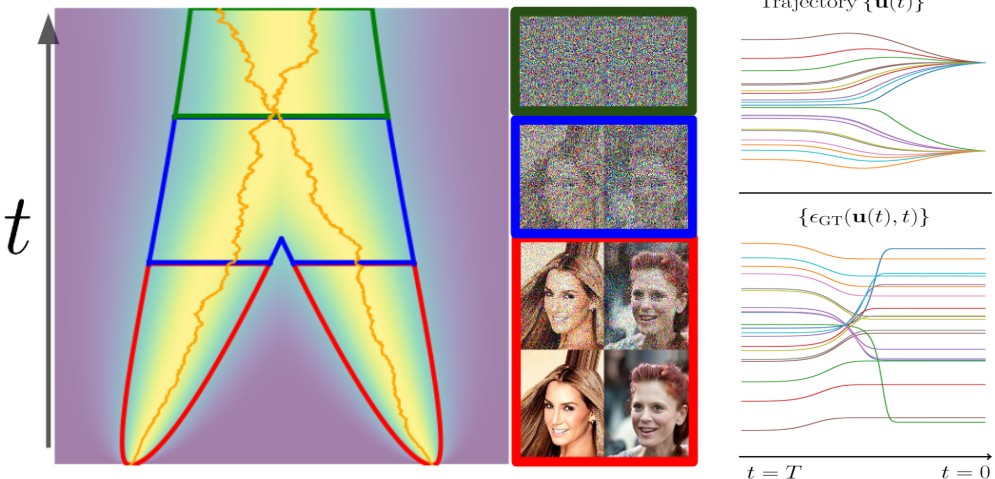

Figure 2: Manifold hypothesis and Dirac distribution assumption. We model an image dataset as a mixture of well-separated Dirac distribution and visualize the diffusion process on the left. Curves in **red** indicate high density area spanned by $p_{0t}(\boldsymbol{u}(t)|\boldsymbol{u}(0))$ by different mode and region surrounded by them indicates the phase when $p_t(\boldsymbol{u})$ is dominated by one mode while region surrounded by **blue** one is for the mixing phase, and **green** region indicates fully mixed phase. On the right, sampling trajectories depict smoothness of $\epsilon_{GT}$ along ODE solutions, which justifies approximations used in DDIM and partially explains its empirical acceleration.

Eq. (6). This approximation Eq. (13) turns out to lead to a numerical scheme for Eq. (6) that coincide with the stochastic DDIM.

**Theorem 1.** *Given the parameterization $\boldsymbol{s}_\theta(\boldsymbol{u}, \tau) = -\frac{\epsilon_\theta(\boldsymbol{u}, \tau)}{\sqrt{1-\alpha_\tau}}$ and the approximation $\boldsymbol{s}_\theta(\boldsymbol{u}, \tau) \approx \frac{1-\alpha_t}{1-\alpha_\tau}\sqrt{\frac{\alpha_\tau}{\alpha_t}}\boldsymbol{s}_\theta(\boldsymbol{u}(t), t) - \frac{1}{1-\alpha_\tau}(\boldsymbol{u} - \sqrt{\frac{\alpha_\tau}{\alpha_t}}\boldsymbol{u}(t))$ for $\tau \in [t - \Delta t, t]$, the exact solution $\boldsymbol{u}(t - \Delta t)$ to Eq. (6) with coefficient Eq. (8) is*

$$\boldsymbol{u}(t - \Delta t) \sim \mathcal{N}\left(\sqrt{\frac{\alpha_{t-\Delta t}}{\alpha_t}}\boldsymbol{u}(t) + \left[-\sqrt{\frac{\alpha_{t-\Delta t}}{\alpha_t}}\sqrt{1-\alpha_t} + \sqrt{1-\alpha_{t-\Delta t} - \sigma_t^2}\right]\epsilon_\theta(\boldsymbol{u}(t), t), \sigma_t^2\boldsymbol{I}_d\right)$$

(14)

*with $\sigma_t = (1 - \alpha_{t-\Delta t})\left[1 - \left(\frac{1-\alpha_{t-\Delta t}}{1-\alpha_t}\right)^{\lambda^2}\left(\frac{\alpha_t}{\alpha_{t-\Delta t}}\right)^{\lambda^2}\right]$, which is the same as the DDIM Eq. (9).*

Note that Thm 1 with $\lambda = 0$ agrees with Prop 2; both reproduce the deterministic DDIM but with different derivations. In summary, DDIMs can be derived by utilizing local approximations.

**Justification of Dirac approximation** While Prop 1 and Prop 3 require the strong assumption that the data distribution is a Dirac, DDIMs in Prop 2 and Thm 1 work very effectively on realistic datasets, which may contain millions of datapoints (Nichol et al., 2021). Here we present one possible interpretation based on the manifold hypothesis (Roweis & Saul, 2000).

It is believed that real-world data lie on a low-dimensional manifold (Tenenbaum et al., 2000) embedded in a high-dimensional space and the data points are well separated in high-dimensional data space. For example, realistic images are scattered in pixel space and the distance between every two images can be very large if measured in pixel difference even if they are similar perceptually. To model this property, we consider a dataset consisting of $M$ datapoints $\{\boldsymbol{u}^{(m)}\}_{m=1}^M$. The exact score is

$$\nabla \log p_t(\boldsymbol{u}) = \sum_m w_m \nabla \log p_{0t}(\boldsymbol{u}|\boldsymbol{u}^{(m)}), \quad w_m = \frac{p_{0t}(\boldsymbol{u}|\boldsymbol{u}^{(m)})}{\sum_m p_{0t}(\boldsymbol{u}|\boldsymbol{u}^{(m)})},$$

(15)

which can be interpreted as a weighted sum of $M$ score functions associated with Dirac distributions. This is illustrated in Fig. 2. In the red color region where the weights $\{w_m\}$ are dominated by one specific data $\boldsymbol{u}^{(m^*)}$ and thus $\nabla \log p_t(\boldsymbol{u}) \approx \nabla \log p_{0t}(\boldsymbol{u}|\boldsymbol{u}^{(m^*)})$. Moreover, in the green region

different modes have similar $\nabla \log p_{0t}(\boldsymbol{u}|\boldsymbol{u}^{(m)})$ as all of them are close to Gaussian and can be approximated by any condition score of any mode. The $\{\epsilon_{\mathrm{GT}}(\boldsymbol{u}(t),t)\}$ trajectories in Fig. 2 validate our hypothesis as we have very smooth curves at the beginning and ending period. The phenomenon that score of realistic datasets can be locally approximated by the score of one datapoint partially justifies the Dirac distribution assumption in Prop 1 and 3 and the effectiveness of DDIMs.

## 4    GENERALIZE AND IMPROVE DDIM

The DDIM is specifically designed for DDPMs. Can we generalize it to other DMs? With the insights in Prop 1 and 3, it turns out that with a carefully chosen $\boldsymbol{K}_\tau$, we can generalize DDIMs to any DMs with general drift and diffusion. We coin the resulted algorithm the *Generalized DDIM (gDDIM)*.

### 4.1    DETERMINISTIC GDDIM WITH PROP 1

**Toy dataset**: Motivated by Prop 1, we ask whether there exists an $\epsilon_{\mathrm{GT}}$ that remains constant along a solution to the probability flow ODE Eq. (7). We start with a special case with initial distribution $p_0(\boldsymbol{u}) = \mathcal{N}(\boldsymbol{u}_0, \Sigma_0)$. It turns out that any solution to Eq. (7) is of the form

$$\boldsymbol{u}(t) = \Psi(t,0)\boldsymbol{u}_0 + \boldsymbol{R}_t\epsilon \tag{16}$$

with a constant $\epsilon$ and a time-varying parameterization coefficients $\boldsymbol{R}_t \in \mathbb{R}^{D \times D}$ that satisfies $\boldsymbol{R}_0 \boldsymbol{R}_0^T = \Sigma_0$ and

$$\frac{d\boldsymbol{R}_t}{dt} = (\boldsymbol{F}_t + \frac{1}{2}\boldsymbol{G}_t \boldsymbol{G}_t^T \Sigma_t^{-1})\boldsymbol{R}_t. \tag{17}$$

Here $\Psi(t,s)$ is the transition matrix associated with $\boldsymbol{F}_\tau$; it is the solution to $\frac{\partial \Psi(t,s)}{\partial t} = \boldsymbol{F}_t \Psi(t,s), \Psi(s,s) = \boldsymbol{I}_D$. Interestingly, $\boldsymbol{R}_t$ satisfies $\boldsymbol{R}_t \boldsymbol{R}_t^T = \Sigma_t$ like $\boldsymbol{K}_t$ in Eq. (4). We remark $\boldsymbol{K}_t = \sqrt{1 - \alpha_t}\boldsymbol{I}_d$ is a solution to Eq. (17) when the DM is specialized to DDPM. Based on Eq. (16) and Eq. (17), we extend Prop 1 to more general DMs.

**Proposition 4.** *Assume the data distribution $p_0(\boldsymbol{u})$ is $\mathcal{N}(\boldsymbol{u}_0, \Sigma_0)$. Let $\boldsymbol{u}(t)$ be an arbitrary solution to the probability flow ODE Eq. (7) with the ground truth score, then $\epsilon_{\mathrm{GT}}(\boldsymbol{u}(t),t) := -\boldsymbol{R}_t^T \nabla \log p_t(\boldsymbol{u}(t))$ remains constant along $\boldsymbol{u}(t)$.*

Note that Prop 4 is slightly more general than Prop 1 in the sense that the initial distribution $p_0$ is a Gaussian instead of a Dirac. Diffusion models with augmented states such as CLD use a Gaussian distribution on the velocity channel for each data point. Thus, when there is a single data point, the initial distribution is a Gaussian instead of a Dirac distribution. A direct consequence of Prop 4 is that we can conduct accurate sampling in one step in the toy example since we can recover the score along any simulated trajectory given its value at $t = T$, if $\boldsymbol{K}_t$ in Eq. (4) is set to be $\boldsymbol{R}_t$. This choice $\boldsymbol{K}_t = \boldsymbol{R}_t$ will make a huge difference in sampling quality as we will show later. The fact provides guidance to design an efficient sampling scheme for realistic data.

**Realistic dataset**: As the accurate score is not available for realistic datasets, we need to use learned score $\boldsymbol{s}_\theta(\boldsymbol{u}, t)$ for sampling. With our new parameterization $\epsilon_\theta(\boldsymbol{u}, t) = -\boldsymbol{R}_t^T \boldsymbol{s}_\theta(\boldsymbol{u}, t)$ and the approximation $\tilde{\epsilon}_\theta(\boldsymbol{u}, \tau) = \epsilon_\theta(\boldsymbol{u}(t), t)$ for $\tau \in [t - \Delta t, t]$, we reach the update step for deterministic gDDIM by solving probability flow with approximator $\tilde{\epsilon}_\theta(\boldsymbol{u}, \tau)$ exactly as

$$\boldsymbol{u}(t - \Delta t) = \Psi(t - \Delta t, t)\boldsymbol{u}(t) + [\int_t^{t-\Delta t} \frac{1}{2}\Psi(t - \Delta t, \tau)\boldsymbol{G}_\tau \boldsymbol{G}_\tau^T \boldsymbol{R}_\tau^{-T} d\tau]\epsilon_\theta(\boldsymbol{u}(t), t), \tag{18}$$

**Multistep predictor-corrector for ODE**: Inspired by Zhang & Chen (2022), we further boost the sampling efficiency of gDDIM by combining Eq. (18) with multistep methods (Hochbruck & Ostermann, 2010; Zhang & Chen, 2022; Liu et al., 2022). We derive multistep *predictor-corrector* methods to reduce the number of steps while retaining accuracy (Press et al., 2007; Sauer, 2005). Empirically, we found that using more NFEs in predictor leads to better performance when the total NFE is small. Thus, we only present multistep predictor for deterministic gDDIM. We include the proof and multistep corrector in App. B. For time discretization grid $\{t_i\}_{i=0}^N$ where $t_0 = 0, t_N = T$,

the $q$-th step predictor from $t_i$ to $t_{i-1}$ in term of $\epsilon_\theta$ parameterization reads

$$\boldsymbol{u}(t_{i-1}) = \Psi(t_{i-1}, t_i)\boldsymbol{u}(t_i) + \sum_{j=0}^{q-1}[\mathbf{C}_{ij}\epsilon_\theta(\boldsymbol{u}(t_{i+j}), t_{i+j})], \tag{19a}$$

$$\mathbf{C}_{ij} = \int_{t_i}^{t_{i-1}} \frac{1}{2}\Psi(t_{i-1}, \tau)\boldsymbol{G}_\tau\boldsymbol{G}_\tau^T\boldsymbol{R}_\tau^{-T}\prod_{k \neq j}[\frac{\tau - t_{i+k}}{t_{i+j} - t_{i+k}}]d\tau. \tag{19b}$$

We note that coefficients in Eqs. (18) and (19b) for general DMs can be calculated efficiently using standard numerical solvers if closed-form solutions are not available.

## 4.2 STOCHASTIC GDDIM WITH PROP 3

Following the same spirits, we generalize Prop 3

**Proposition 5.** *Assume the data distribution $p_0(\boldsymbol{u})$ is $\mathcal{N}(\boldsymbol{u}_0, \Sigma_0)$. Given any evaluation of the score function $\nabla \log p_s(\boldsymbol{u}(s))$, one can recover $\nabla \log p_t(\boldsymbol{u})$ for any $t, \boldsymbol{u}$ as*

$$\nabla \log p_t(\boldsymbol{u}) = \Sigma_t^{-1}\Psi(t, s)\Sigma_s\nabla \log p_s(\boldsymbol{u}(s)) - \Sigma_t^{-1}[\boldsymbol{u} - \Psi(t, s)\boldsymbol{u}(s)]. \tag{20}$$

Prop 5 is not surprising; in our example, the score has a closed form. Eq. (20) not only provides an accurate score estimation for our toy dataset, but also serves as a score approximator for realistic data.

**Realistic dataset**: Based on Eq. (20), with the parameterization $\boldsymbol{s}_\theta(\boldsymbol{u}, \tau) = -\boldsymbol{R}_\tau^{-T}\epsilon_\theta(\boldsymbol{u}, \tau)$, we propose the following gDDIM approximator $\tilde{\epsilon}_\theta(\boldsymbol{u}, \tau)$ for $\epsilon_\theta(\boldsymbol{u}, \tau)$

$$\tilde{\epsilon}_\theta(\boldsymbol{u}, \tau) = \boldsymbol{R}_\tau^{-1}\Psi(\tau, s)\boldsymbol{R}_s\epsilon_\theta(\boldsymbol{u}(s), s) + \boldsymbol{R}_\tau^{-1}[\boldsymbol{u} - \Psi(\tau, s)\boldsymbol{u}(s)]. \tag{21}$$

**Proposition 6.** *With the parameterization $\epsilon_\theta(\boldsymbol{u}, t) = -\boldsymbol{R}_t^T\boldsymbol{s}_\theta(\boldsymbol{u}, t)$ and the approximator $\tilde{\epsilon}_\theta(\boldsymbol{u}, \tau)$ in Eq. (21), the solution to Eq. (6) satisfies*

$$\boldsymbol{u}(t) \sim \mathcal{N}(\Psi(t, s)\boldsymbol{u}(s) + [\hat{\Psi}(t, s) - \Psi(t, s)]\boldsymbol{R}_s\epsilon_\theta(\boldsymbol{u}(s), s), \boldsymbol{P}_{st}), \tag{22}$$

*where $\hat{\Psi}(t, s)$ is the transition matrix associated with $\hat{\boldsymbol{F}}_\tau := \boldsymbol{F}_\tau + \frac{1+\lambda^2}{2}\boldsymbol{G}_\tau\boldsymbol{G}_\tau^T\Sigma_\tau^{-1}$ and the covariance matrix $\boldsymbol{P}_{st}$ solves*

$$\frac{d\boldsymbol{P}_{s\tau}}{d\tau} = \hat{\boldsymbol{F}}_\tau\boldsymbol{P}_{s\tau} + \boldsymbol{P}_{s\tau}\hat{\boldsymbol{F}}_\tau^T + \lambda^2\boldsymbol{G}_\tau\boldsymbol{G}_\tau^T, \quad \boldsymbol{P}_{ss} = 0. \tag{23}$$

Our stochastic gDDIM then uses Eq. (22) for update. Though the stochastic gDDIM and the deterministic gDDIM look quite different from each other, there exists a connection between them.

**Proposition 7.** *Eq. (22) in stochastic gDDIM reduces to Eq. (18) in deterministic gDDIM when $\lambda = 0$.*

## 5 EXPERIMENTS

As gDDIM reduces to DDIM for VPSDE and DDIM proves very successful, we validate the generation and effectiveness of gDDIM on CLD and BDM. We design experiments to answer the following questions. How to verify Prop 4 and 5 empirically? Can gDDIM improve sampling efficiency compared with existing works? What differences do the choice of $\lambda$ and $\boldsymbol{K}_t$ make? We conduct experiments with different DMs and sampling algorithms on CIFAR10 for quantitative comparison. We include more illustrative experiments on toy datasets, high dimensional image datasets, and more baseline comparison in App. C.

**Choice of $\boldsymbol{K}_t$:** A key of gDDIM is the special choice $\boldsymbol{K}_t = \boldsymbol{R}_t$ which is obtained via solving Eq. (17). In CLD, Dockhorn et al. (2021) choose $\boldsymbol{K}_t = \boldsymbol{L}_t$ based on Cholesky decomposition of $\Sigma_t$ and it does not obey Eq. (17). More details regarding $\boldsymbol{L}_t$ are included in App. C. As it is shown in Fig. 1, on real datasets with a trained score model, we randomly pick pixel locations and check the pixel value and $\epsilon_\theta$ output along the solutions to the probability flow ODE produced by the high-resolution ODE solver. With the choice $\boldsymbol{K}_t = \boldsymbol{L}_t$, $\epsilon_\theta^{(L)}(\boldsymbol{u}, t; \boldsymbol{v})$ suffers from oscillation like $\boldsymbol{x}$

value along time. However, $\epsilon_\theta^{(\boldsymbol{R})}(\boldsymbol{u}, t)$ is much more flat. We further compare samples generated by $\boldsymbol{L}_t$ and $\boldsymbol{R}_t$ parameterizaiton in Tab. 1, where both use the multistep exponential solver in Eq. (19).

Table 1: $\boldsymbol{L}_t$ vs $\boldsymbol{R}_t$ (Our) on CLD

| | FID at different NFE | | | |
|---|---|---|---|---|
| $\boldsymbol{K}_t$ | 20 | 30 | 40 | 50 |
| $\boldsymbol{L}_t$ | 368 | 167 | 4.12 | 3.31 |
| $\boldsymbol{R}_t$ | 3.90 | 2.64 | 2.37 | 2.26 |

Table 2: $\lambda$ and integrators choice with NFE=50

| | FID at different $\lambda$ | | | | | |
|---|---|---|---|---|---|---|
| Method | 0.0 | 0.1 | 0.3 | 0.5 | 0.7 | 1.0 |
| gDDIM | 5.17 | 5.51 | 12.13 | 33 | 41 | 49 |
| EM | 346 | 168 | 137 | 89 | 45 | 57 |

**Choice of $\lambda$:** We further conduct a study with different $\lambda$ values. Note that polynomial extrapolation in Eq. (19) is not used here even when $\lambda = 0$. As it is shown in Tab. 2, increasing $\lambda$ deteriorates the sample quality, demonstrating our claim that deterministic DDIM has better performance than its stochastic counterpart when a small NFE is used. We also find stochastic gDDIM significantly outperforms EM, which indicates the effectiveness of the approximation Eq. (21).

**Accelerate various DMs:** We present a comparison among various DMs and various sampling algorithms. To make a fair comparison, we compare three DMs with similar size networks while retaining other hyperparameters from their original works. We make two modifications to DDPM, including continuous-time training (Song et al., 2020b) and smaller stop sampling time (Karras et al., 2022), which help improve sampling quality empirically. For BDM, we note Hoogeboom & Salimans (2022) only supports the ancestral sampling algorithm, a variant of EM algorithm. With reformulated noising and denoising process as SDE Eq. (11), we can generate samples by solving corresponding SDE/ODEs. The sampling quality of gDDIM with 50 NFE can outperform the original ancestral sampler with 1000 NFE, more than 20 times acceleration.

Table 3: Acceleration on various DMs with similar training pipelines and architecture. For RK45, we tune its tolerance hyperparameters so that the real NFE is close but not equal to the given NFE. †: pre-trained model from Song et al. (2020b). ††: Karras et al. (2022) apply Heun method in rescaled DM, which is essentially a variant of DEIS (Zhang & Chen, 2022)

| DM | Sampler | FID ($\downarrow$) under different NFE | | | | |
|---|---|---|---|---|---|---|
| | | 10 | 20 | 50 | 100 | 1000 |
| **DDPM**[†] | EM | >100 | >100 | 31.2 | 12.2 | 2.64 |
| | Prob.Flow, RK45 | >100 | 52.5 | 6.62 | 2.63 | **2.56** |
| | 2$^{\text{nd}}$ Heun[††] | 66.25 | 6.62 | 2.65 | 2.57 | **2.56** |
| | gDDIM | **4.17** | **3.03** | **2.59** | **2.56** | **2.56** |
| **BDM** | Ancestral sampling | >100 | >100 | 29.8 | 9.73 | 2.51 |
| | Prob.Flow, RK45 | >100 | 68.2 | 7.12 | 2.58 | **2.46** |
| | gDDIM | **4.52** | **2.97** | **2.49** | **2.47** | **2.46** |
| **CLD** | EM | >100 | >100 | 57.72 | 13.21 | 2.39 |
| | Prob.Flow, RK45 | >100 | >100 | 31.7 | 4.56 | **2.25** |
| | gDDIM | **13.41** | **3.39** | **2.26** | **2.26** | **2.25** |

## 6 CONCLUSIONS AND LIMITATIONS

**Contribution**: The more structural knowledge we leverage, the more efficient algorithms we obtain. In this work, we provide a clean interpretation of DDIMs based on the manifold hypothesis and the sparsity property on realistic datasets. This new perspective unboxes the numerical discretization used in DDIM and explains the advantage of ODE-based sampler over SDE-based when NFE is small. Based on this interpretation, we extend DDIMs to general diffusion models. The new algorithm, gDDIM, only requires a tiny but elegant modification to the parameterization of the score model and improves sampling efficiency drastically. We conduct extensive experiments to validate the effectiveness of our new sampling algorithm. **Limitation**: There are several promising future directions. First, though gDDIM is designed for general DMs, we only verify it on three DMs. It is beneficial to explore more efficient diffusion processes for different datasets, in which we believe gDDIM will play an important role in designing sampling algorithms. Second, more investigations are needed to design an efficient sampling algorithm by exploiting more structural knowledge in DMs. The structural knowledge can originate from different sources such as different modalities of datasets, and mathematical structures presented in specific diffusion processes.

ACKNOWLEDGMENTS

The authors would like to thank the anonymous reviewers for their useful comments. This work is partially supported by NSF ECCS-1942523, NSF CCF-2008513, and NSF DMS-1847802.

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

## A  More related works

Learning generative models with DMs via score matching has received tremendous attention recently (Sohl-Dickstein et al., 2015; Lyu, 2012; Song & Ermon, 2019; Song et al., 2020b; Ho et al., 2020; Nichol & Dhariwal, 2021). However, the sampling efficiency of DMs is still not satisfying. Jolicoeur-Martineau et al. (2021a) introduced adaptive solvers for SDEs associated with DMs for the task of image generation. Song et al. (2020a) modified the forward noising process into a non-Markov process without changing the training objective function. The authors then proposed a family of samplers, including deterministic DDIM and stochastic DDIM, based on the modifications. Both of the samplers demonstrate significant improvements over previous samplers. There are variants of the DDIM that aim to further improve the sampling quality and efficiency. Deterministic DDIM in fact reduces to probability flow in the infinitesimal step size limit (Song et al., 2020a; Liu et al., 2022). Meanwhile, various approaches have been proposed to accelerate DDIM (Kong & Ping, 2021b; Watson et al., 2021; Liu et al., 2022). Bao et al. (2022) improved the DDIM by optimizing the reverse variance in DMs. Watson et al. (2022) generalized the DDIM in DDPM with learned update coefficients, which are trained by minimizing an external perceptual loss. Nichol & Dhariwal (2021) tuned the variance of the schedule of DDPM. Liu et al. (2022) found that the DDIM is a pseudo numerical method and proposed a pseudo linear multi-step method for it. Zhang & Chen (2022) discovered that DDIMs are numerical integrators for marginal-equivalent SDEs, and the deterministic DDIM is actually an exponential integrator for the probability flow ODE. They further utilized exponential multistep methods to boost sampling performance for VPSDE.

Another promising approach to accelerate the diffusion model is distillation for the probability flow ODE. Luhman & Luhman (2021) proposed to learn the map from noise to data in a teacher-student fashion, where supervised signals are provided by simulating the deterministic DDIM. The final student network distilled is able to generate samples with reasonable quality within one step. Salimans & Ho (2022) proposed a new progressive distillation approach to improve training efficiency and stability. This distillation approach relies on solving the probability flow ODE and needs extra training procedures. Since we generalize and improve the DDIM in this work, it will be beneficial to combine this distillation method with our algorithm for better performance in the future.

Recently, numerical structures of DMs have received more and more attention; they play important roles in efficient sampling methods. Dockhorn et al. (2021) designed Symmetric Splitting CLD Sampler (SSCS) that takes advantage of Hamiltonian structure of the CLD and demonstrated advantages over the naive Euler-Maruyama method. Zhang & Chen (2022) first utilized the semilinear structure presented in DMs and showed that the exponential integrator gave much better sampling quality than the Euler method. The proposed Diffusion Exponential Integrator Sampler (DEIS) further accelerates sampling by utilizing Multistep and Runge Kutta ODE solvers. Similar to DEIS, Lu et al. (2022); Karras et al. (2022) also proposed ODE solvers that utilize analytical forms of diffuson scheduling coefficients. Karras et al. (2022) further improved the stochastic sampling algorithm by augmenting ODE trajectories with small noise. Further tailoring these integrators to account for the stiff property of ODEs (Hochbruck & Ostermann, 2010; Whalen et al., 2015) is a promising direction in fast sampling for DMs.

## B  Proofs

Since the proposed gDDIM is a generalization of DDIM, the results regarding gDDIM in Sec. 4 are generalizations of those in Sec. 3. In particular, Prop 4 generalizes Prop 1, Eq. (18) generalizes Prop 2, and Prop 5 generalizes Prop 3. Thus, for the sake of simplicity, we mainly present proofs for gDDIM in Sec. 4.

### B.1  Blurring Diffusion Models

We first review the formulations regarding BDM proposed in Hoogeboom & Salimans (2022) and show it can be reformulated as SDE in continuous time Eq. (11).

Hoogeboom & Salimans (2022) introduce an forwarding noising scheme where noise corrupts data in frequency space with different schedules for the dimensions. Different from existing DMs, the

diffusion process is defined in frequency space:

$$p(\boldsymbol{y}_t|\boldsymbol{y}_0) = \mathcal{N}(\boldsymbol{y}_t|\boldsymbol{\alpha}_t\boldsymbol{y}_0, \boldsymbol{\sigma}_t\boldsymbol{I}) \quad \boldsymbol{y}_t = \boldsymbol{V}^T\boldsymbol{x}_t \tag{24}$$

where $\boldsymbol{\alpha}, \boldsymbol{\sigma}$ are $\mathbb{R}^{d\times d}$ diagonal matries and control the different diffuse rate for data along different dimension. And $\boldsymbol{y}_t$ is the mapping of $\boldsymbol{x}_t$ in frequency domain obtained through Discrete Cosine Transform (DCT) $\boldsymbol{V}^T$. We note $\boldsymbol{V}$ is the inverse DCT mapping and $\boldsymbol{V}^T\boldsymbol{V} = \boldsymbol{I}$.

Based on Eq. (24), we are able to derive its corresponding noising scheme in original data space

$$p(\boldsymbol{x}_t|\boldsymbol{x}_0) = \mathcal{N}(\boldsymbol{x}_t|\boldsymbol{V}\boldsymbol{\alpha}_t\boldsymbol{V}^T\boldsymbol{x}_0, \boldsymbol{\sigma}_t\boldsymbol{I}). \tag{25}$$

Eq. (25) indicates BDM is a non-isotropic diffusion process. Therefore, we are able to derive its forward process as a linear SDE. For a general linear SDE Eq. (1), its mean and covariance follow

$$\frac{d\boldsymbol{m}_t}{dt} = \boldsymbol{F}_t\boldsymbol{m}_t \tag{26}$$

$$\frac{d\Sigma_t}{dt} = \boldsymbol{F}_t\Sigma_t + \Sigma_t\boldsymbol{F}_t + \boldsymbol{G}_t\boldsymbol{G}_t^T. \tag{27}$$

Plugging Eq. (25) into Eq. (26), we are able to derive the drift and diffusion $\boldsymbol{F}_t, \boldsymbol{G}_t$ for BDM as Eq. (11). As BDM admit a SDE formulation, we can use Eq. (3) to train BDM. For choice of hyperparameters $\boldsymbol{\alpha}_t, \boldsymbol{\sigma}_t$ and pratical implementation, we include more details in App. C.

## B.2 DETERMINISTIC GDDIM

### B.2.1 PROOF OF EQ. (17)

Since we assume data distribution $p_0(\boldsymbol{u}) = \mathcal{N}(\boldsymbol{u}_0, \Sigma_0)$, the score has closed form

$$\nabla \log p_t(\boldsymbol{u}) = -\Sigma_t^{-1}(\boldsymbol{u} - \Psi(t,0)\boldsymbol{u}_0). \tag{28}$$

To make sure our construction Eq. (16) is a solution to the probability flow ODE, we examine the condition for $\boldsymbol{R}_t$. The LHS of the probability flow ODE is

$$\begin{aligned} d\boldsymbol{u} &= d[\Psi(t,0)\boldsymbol{u}_0 + \boldsymbol{R}_t\epsilon] \\ &= \dot{\Psi}(t,0)\boldsymbol{u}_0 dt + \dot{\boldsymbol{R}}_t\epsilon dt \\ &= [\boldsymbol{F}_t\Psi(t,0)\boldsymbol{u}_0 + \dot{\boldsymbol{R}}_t\epsilon]dt. \end{aligned} \tag{29}$$

The RHS of the probability flow ODE is

$$\begin{aligned} [\boldsymbol{F}_t\boldsymbol{u} - \frac{1}{2}\boldsymbol{G}_t\boldsymbol{G}_t^T\nabla \log p_t(\boldsymbol{u})]dt &= [\boldsymbol{F}_t\Psi(t,0)\boldsymbol{u}_0 + \boldsymbol{F}_t\boldsymbol{R}_t\epsilon + \frac{1}{2}\boldsymbol{G}_t\boldsymbol{G}_t^T\boldsymbol{R}_t^{-T}\epsilon]dt \\ &= [\boldsymbol{F}_t\Psi(t,0)\boldsymbol{u}_0 + \boldsymbol{F}_t\boldsymbol{R}_t\epsilon + \frac{1}{2}\boldsymbol{G}_t\boldsymbol{G}_t^T\boldsymbol{R}_t^{-T}\boldsymbol{R}_t^{-1}\boldsymbol{R}_t\epsilon]dt, \end{aligned} \tag{30}$$

where the first equality is due to $\nabla \log p_t(\boldsymbol{u}) = -\boldsymbol{R}_t^{-T}\epsilon$.

Since Eqs. (29) and (30) holds for each $\epsilon$, we establish

$$\dot{\boldsymbol{R}}_t = (\boldsymbol{F}_t + \frac{1}{2}\boldsymbol{G}_t\boldsymbol{G}_t^T\boldsymbol{R}_t^{-T}\boldsymbol{R}_t^{-1})\boldsymbol{R}_t \tag{31}$$

$$= (\boldsymbol{F}_t + \frac{1}{2}\boldsymbol{G}_t\boldsymbol{G}_t^T\Sigma_t^{-1})\boldsymbol{R}_t. \tag{32}$$

### B.2.2 PROOF OF PROP 4

Similar to the proof of Eq. (17), over a solution $\{\boldsymbol{u}(t), t\}$ to the probability flow ODE, $\boldsymbol{R}_t^{-1}(\boldsymbol{u}(t) - \Psi(t,0)\boldsymbol{u}_0)$ is constant. Furthermore, by Eq. (28),

$$\begin{aligned} \nabla \log p_t(\boldsymbol{u}(t)) &= -\Sigma_t^{-1}(\boldsymbol{u}(t) - \Psi(t,0)\boldsymbol{u}_0) \\ &= -\boldsymbol{R}_t^{-T}\boldsymbol{R}_t^{-1}(\boldsymbol{u}(t) - \Psi(t,0)\boldsymbol{u}_0) \end{aligned} \tag{33}$$

Eq. (33) implies that $-\boldsymbol{R}_t^T\nabla \log p_t(\boldsymbol{u}(t))$ is a constant and invariant with respect to $t$.

### B.2.3 PROOF OF EQS. (12) AND (18)

We derive Eq. (18) first.

The update step is based on the approximation $\tilde{\epsilon}_\theta(\boldsymbol{u}, \tau) = \epsilon_\theta(\boldsymbol{u}(t), t)$ for $\tau \in [t - \Delta t, t]$. The resultant ODE with $\tilde{\epsilon}_\theta$ reads

$$\dot{\boldsymbol{u}} = \boldsymbol{F}_\tau \boldsymbol{u} + \frac{1}{2}\boldsymbol{G}_\tau \boldsymbol{G}_\tau^T \boldsymbol{R}_\tau^{-1} \epsilon_\theta(\boldsymbol{u}(t), t), \tag{34}$$

which is a linear ODE. The closed-form solution reads

$$\boldsymbol{u}(t_{i-1}) = \Psi(t_{i-1}, t_i)\boldsymbol{u}(t) + [\int_{t_i}^{t_{i-1}} \frac{1}{2}\Psi(t_{i-1}, \tau)\boldsymbol{G}_\tau \boldsymbol{G}_\tau^T \boldsymbol{R}_\tau^{-T} d\tau]\epsilon_\theta(\boldsymbol{u}(t), t), \tag{35}$$

where $\Psi(t, s)$ is the transition matrix associated $\boldsymbol{F}_t$, that is, $\Psi$ satisfies

$$\frac{d\Psi(t, s)}{dt} = \boldsymbol{F}_t \Psi(t, s) \quad \Psi(s, s) = \boldsymbol{I}_D. \tag{36}$$

When the DM is specified to be DDPM, we derive Eq. (12) based on Eq. (18) by expanding the coefficients in Eq. (18) explicitly as

$$\Psi(t, s) = \sqrt{\frac{\alpha_t}{\alpha_s}},$$

$$\Psi(t - \Delta t, t) = \sqrt{\frac{\alpha_{t-\Delta t}}{\alpha_t}},$$

$$\int_t^{t-\Delta t} \frac{1}{2}\Psi(t - \Delta t, \tau)\boldsymbol{G}_\tau \boldsymbol{G}_\tau^T \boldsymbol{R}_\tau^{-1} d\tau = \int_t^{t-\Delta t} -\frac{1}{2}\sqrt{\frac{\alpha_{t-\Delta t}}{\alpha_\tau}}\frac{d\log\alpha_\tau}{d\tau}\frac{1}{\sqrt{1 - \alpha_\tau}}d\tau$$

$$= \sqrt{\alpha_{t-\Delta t}}\sqrt{\frac{1 - \alpha_\tau}{\alpha_\tau}}\Big|_{\alpha_t}^{\alpha_{t-\Delta t}}$$

$$= \sqrt{1 - \alpha_{t-\Delta t}} - \sqrt{1 - \alpha_t}\sqrt{\frac{\alpha_{t-\Delta t}}{\alpha_t}}.$$

### B.2.4 PROOF OF MULTISTEP PREDICTOR-CORRECTOR

Our Multistep Predictor-Corrector method slightly extends the traditional linear multistep Predictor-Corrector method to incorporate the semilinear structure in the probability flow ODE with an exponential integrator (Press et al., 2007; Hochbruck & Ostermann, 2010).

**Predictor:**

For Eq. (7), the key insight of the multistep predictor is to use existing function evaluations $\epsilon_\theta(\boldsymbol{u}(t_i), t_i), \epsilon_\theta(\boldsymbol{u}(t_{i+1}), t_{i+1}), \cdots, \epsilon_\theta(\boldsymbol{u}(t_{i+q-1}), t_{i+q-1})$ and their timestamps $t_i, t_{i+1}, \cdots, t_{i+q-1}$ to fit a $q - 1$ order polynomial $\epsilon_p(t)$ to approximate $\epsilon_\theta(\boldsymbol{u}(\tau), \tau)$. With this approximator $\tilde{\epsilon}_\theta(\boldsymbol{u}, \tau) = \epsilon_p(\tau)$ for $\tau \in [t_{i-1}, t_i]$, the multistep predictor step is obtained by solving

$$\frac{d\boldsymbol{u}}{dt} = \boldsymbol{F}_t \boldsymbol{u} + \frac{1}{2}\boldsymbol{G}_\tau \boldsymbol{G}_\tau^T \boldsymbol{R}_\tau^{-T}\tilde{\epsilon}_\theta(\boldsymbol{u}, \tau)$$

$$= \boldsymbol{F}_t \boldsymbol{u} + \frac{1}{2}\boldsymbol{G}_\tau \boldsymbol{G}_\tau^T \boldsymbol{R}_\tau^{-T}\epsilon_p(\tau), \tag{37}$$

which is a linear ODE. The solution to Eq. (37) satisfies

$$\boldsymbol{u}(t_{i-1}) = \Psi(t_{i-1}, t_i)\boldsymbol{u}(t_i) + \int_{t_i}^{t_{i-1}} \frac{1}{2}\boldsymbol{G}_\tau \boldsymbol{G}_\tau^T \boldsymbol{R}_\tau^{-T}\epsilon_p(\tau)d\tau. \tag{38}$$

Based on Lagrange formula, we can write $\epsilon_p(\tau)$ as

$$\epsilon_p(\tau) = \sum_{j=0}^{q-1}[\prod_{k\neq j} \frac{\tau - t_{i+k}}{t_{i+j} - t_{i+k}}]\epsilon_\theta(\boldsymbol{u}_{t_{i+j}}, t_{i+j}). \tag{39}$$

Plugging Eq. (39) into Eq. (38), we obtain

$$\boldsymbol{u}(t_{i-1}) = \Psi(t_{i-1}, t_i)\boldsymbol{u}(t_i) + \sum_{j=0}^{q-1} [{}^{p}\mathrm{C}_{ij}^{(q)}\epsilon_\theta(\boldsymbol{u}(t_{i+j}), t_{i+j})], \tag{40}$$

$$^{p}\mathrm{C}_{ij}^{(q)} = \int_{t_i}^{t_{i-1}} \frac{1}{2}\Psi(t_{i-1}, \tau)\boldsymbol{G}_\tau\boldsymbol{G}_\tau^T\boldsymbol{R}_\tau^{-T} \prod_{k\neq j, k=0}^{q-1} [\frac{\tau - t_{i+k}}{t_{i+j} - t_{i+k}}]d\tau, \tag{41}$$

which are Eqs. (19a) and (19b). Here we use ${}^{p}\mathrm{C}_{ij}^{(q)}$ to emphasize these are constants used in the $q$-step predictor. The 1-step predictor reduces to Eq. (18).

**Corrector:**

Compared with the explicit scheme for the multistep predictor, the multistep corrector behaves like an implicit method (Press et al., 2007). Instead of constructing $\epsilon_p(\tau)$ to extrapolate model output for $\tau \in [t_{i-1}, t_i]$ as in the predictor, the $q$ step corrector aims to find $\epsilon_c(\tau)$ to interpolate $\epsilon_\theta(\boldsymbol{u}(t_{i-1}), t_{i-1}), \epsilon_\theta(\boldsymbol{u}(t_i), t_i), \epsilon_\theta(\boldsymbol{u}(t_{i+1}), t_{i+1}), \cdots, \epsilon_\theta(\boldsymbol{u}(t_{i+q-2}), t_{i+q-2})$ and their timestamps $t_{i-1}, t_i, t_{i+1}, \cdots, t_{i+q-2}$. Thus, $\boldsymbol{u}(t_{i-1})$ is obtained by solving

$$\boldsymbol{u}(t_{i-1}) = \Psi(t_{i-1}, t_i)\boldsymbol{u}(t_i) + \int_{t_i}^{t_{i-1}} \frac{1}{2}\boldsymbol{G}_\tau\boldsymbol{G}_\tau^T\boldsymbol{R}_\tau^{-T}\epsilon_c(\tau)d\tau. \tag{42}$$

Since $\epsilon_c(\tau)$ is defined implicitly, it is not easy to find $\epsilon_c(\tau), \boldsymbol{u}(t_{i-1})$. Instead, practitioners bypass the difficulties by interpolating $\epsilon_\theta(\bar{\boldsymbol{u}}(t_{i-1}), t_{i-1}), \epsilon_\theta(\bar{\boldsymbol{u}}(t_i), t_i), \epsilon_\theta(\bar{\boldsymbol{u}}(t_{i+1}), t_{i+1}), \cdots, \epsilon_\theta(\bar{\boldsymbol{u}}(t_{i+q-2}), t_{i+q-2})$ where $\bar{\boldsymbol{u}}(t_{i-1})$ is obtained by the predictor in Eq. (38) and $\bar{\boldsymbol{u}}(t_i) = \boldsymbol{u}(t_i), \bar{\boldsymbol{u}}(t_{i+1}) = \boldsymbol{u}(t_{i+1}), \cdots, \bar{\boldsymbol{u}}(t_{i+q-2}) = \boldsymbol{u}(t_{i+q-2})$. Hence, we derive the update step for corrector based on

$$\boldsymbol{u}(t_{i-1}) = \Psi(t_{i-1}, t_i)\boldsymbol{u}(t_i) + \int_{t_i}^{t_{i-1}} \frac{1}{2}\boldsymbol{G}_\tau\boldsymbol{G}_\tau^T\boldsymbol{R}_\tau^{-T}\epsilon_c(\tau)d\tau, \tag{43}$$

where $\epsilon_c(\tau)$ is defined as

$$\epsilon_c(\tau) = \sum_{j=-1}^{q-2} [\prod_{k\neq j} \frac{\tau - t_{i+k}}{t_{i+j} - t_{i+k}}]\epsilon_\theta(\bar{\boldsymbol{u}}_{t_{i+j}}, t_{i+j}). \tag{44}$$

Plugging Eq. (44) into Eq. (43), we reach the update step for the corrector

$$\boldsymbol{u}(t_{i-1}) = \Psi(t_{i-1}, t_i)\boldsymbol{u}(t_i) + \sum_{j=-1}^{q-2} [{}^{c}\mathrm{C}_{ij}^{(q)}\epsilon_\theta(\bar{\boldsymbol{u}}(t_{i+j}), t_{i+j})], \tag{45}$$

$$^{c}\mathrm{C}_{ij}^{(q)} = \int_{t_i}^{t_{i-1}} \frac{1}{2}\Psi(t_{i-1}, \tau)\boldsymbol{G}_\tau\boldsymbol{G}_\tau^T\boldsymbol{R}_\tau^{-T} \prod_{k\neq j, k=-1}^{q-2} [\frac{\tau - t_{i+k}}{t_{i+j} - t_{i+k}}]d\tau. \tag{46}$$

We use ${}^{c}\mathrm{C}_{ij}^{(q)}$ to emphasis constants used in the $q$-step corrector.

**Exponential multistep Predictor-Corrector:**

Here we present the Exponential multistep Predictor-Corrector algorithm. Specifically, we employ one $q$-step corrector update step after an update step of the $q$-step predictor. The interested reader can easily extend the idea to employ multiple update steps of corrector or different number of steps for the predictor and the corrector. We note coefficients ${}^{p}\mathrm{C}, {}^{c}\mathrm{C}$ can be calculated using high resolution ODE solver once and used everywhere.

---

**Algorithm 1** Exponential multistep Predictor-Corrector

---

**Input**: Timestamps $\{t_i\}_{i=0}^N$, step order $q$, coefficients for predictor update $^p\mathrm{C}$, coefficients for corrector update $^c\mathrm{C}$
**Instantiate**: $\boldsymbol{u}(t_N) \sim p_T(\boldsymbol{u})$
**for** $i$ in $N, N-1, \cdots, 1$ **do**
   # predictor update step
   $q_{\mathrm{cur}} = \min(q, N-i+1)$ # handle warming start, use lower order multistep method
   $\boldsymbol{u}_{t_{i-1}} \leftarrow$ Simulate Eq. (40) with $q_{\mathrm{cur}}$-step predictor
   # corrector update step
   $q_{\mathrm{cur}} = \min(q, N-i+2)$ # handle warming start, use lower order multistep method
   $\bar{\boldsymbol{u}}(t_{i-1}), \bar{\boldsymbol{u}}(t_i), \cdots, \bar{\boldsymbol{u}}(t_{i+q_{\mathrm{cur}}-1}) \leftarrow \boldsymbol{u}(t_{i-1}), \boldsymbol{u}(t_i), \cdots, \boldsymbol{u}(t_{i+q_{\mathrm{cur}}-1})$
   $\boldsymbol{u}_{t_{i-1}} \leftarrow$ Simulate Eq. (45) with $q_{\mathrm{cur}}$-step corrector
**end for**

---

### B.3 STOCHASTIC GDDIM

#### B.3.1 PROOF OF PROP 5

Assuming that the data distribution $p_0(\boldsymbol{u})$ is $\mathcal{N}(\boldsymbol{u}_0, \Sigma_0)$ with a given $\Sigma_0$, we can derive the mean and covariance of $p_t(\boldsymbol{u})$ as

$$\mu_t = \Psi(t, 0) \tag{47}$$

$$\frac{d\Sigma_t}{dt} = \boldsymbol{F}_t \Sigma_t + \Sigma_t \boldsymbol{F}_t^T + \boldsymbol{G}_t \boldsymbol{G}_t^T. \tag{48}$$

Therefore, the ground truth score reads

$$\nabla \log p_t(\boldsymbol{u}) = -\Sigma_t^{-1}(\boldsymbol{u} - \Psi(t, 0)\boldsymbol{u}_0). \tag{49}$$

We assume $\Sigma_0$ is given but $\boldsymbol{u}_0$ is unknown. Fortunately, $\boldsymbol{u}_0$ can be inferred via one score evaluation as follows. Given evaluation $\nabla \log p_s(\boldsymbol{u}(s))$, we can recover $\boldsymbol{u}_0$ as

$$\boldsymbol{u}_0 = \Psi(0, s)[\Sigma_s \nabla \log p_s(\boldsymbol{u}(s)) + \boldsymbol{u}(s)]. \tag{50}$$

Plugging Eq. (50) and $\Psi(t, s) = \Psi(t, 0)\Psi(0, s)$ into Eq. (49), we recover Eq. (20).

#### B.3.2 PROOF OF PROP 6

With the approximator $\tilde{\epsilon}_\theta(\boldsymbol{u}, t)$ defined in Eq. (21) for $\tau \in [s, t]$, Eq. (6) can be reformulated as

$$d\boldsymbol{u} = \boldsymbol{F}_\tau \boldsymbol{u} d\tau + \frac{1+\lambda^2}{2} \boldsymbol{G}_\tau \boldsymbol{G}_\tau^T \boldsymbol{R}_\tau^{-T} \tilde{\epsilon}_\theta(\boldsymbol{u}, \tau) dt + \lambda \boldsymbol{G}_\tau d\boldsymbol{w}$$

$$= (\boldsymbol{F}_\tau + \frac{1+\lambda^2}{2} \boldsymbol{G}_\tau \boldsymbol{G}_\tau^T \boldsymbol{R}_\tau^{-T} \boldsymbol{R}_\tau^{-1}) \boldsymbol{u} dt$$

$$+ \frac{1+\lambda^2}{2} \boldsymbol{G}_\tau \boldsymbol{G}_\tau^T \boldsymbol{R}_\tau^{-T} \boldsymbol{R}_\tau^{-1} \Psi(\tau, s)(\boldsymbol{R}_s \epsilon_\theta(\boldsymbol{u}(s), s) - \boldsymbol{u}(s)) dt + \lambda \boldsymbol{G}_\tau d\boldsymbol{w}. \tag{51}$$

Define $\hat{\boldsymbol{F}}_\tau := \boldsymbol{F}_\tau + \frac{1+\lambda^2}{2} \boldsymbol{G}_\tau \boldsymbol{G}_\tau^T \boldsymbol{R}_\tau^{-T} \boldsymbol{R}_\tau^{-1} = \boldsymbol{F}_\tau + \frac{1+\lambda^2}{2} \boldsymbol{G}_\tau \boldsymbol{G}_\tau^T \Sigma_\tau^{-1}$, and denote by $\hat{\Psi}(t, s)$ the transition matrix associated with it. Clearly, Eq. (51) is a linear differential equation on $\boldsymbol{u}$, and the conditional probability $\hat{p}_{st}(\boldsymbol{u}(t)|\boldsymbol{u}(s))$ associated with it is a Gaussian distribution.

Applying Särkkä & Solin (2019, Eq (6.6,6.7)), we obtain the exact expressions

$$\text{Mean} = \hat{\Psi}(t, s)\boldsymbol{u}(s) - [\int_s^t \hat{\Psi}(t, \tau) \frac{1+\lambda^2}{2} \boldsymbol{G}_\tau \boldsymbol{G}_\tau^T \Sigma_\tau^{-1} \Psi(\tau, s)] d\tau \, \boldsymbol{u}(s)$$

$$+ [\int_s^t \hat{\Psi}(t, \tau) \frac{1+\lambda^2}{2} \boldsymbol{G}_\tau \boldsymbol{G}_\tau^T \Sigma_\tau^{-1} \Psi(\tau, s)] d\tau \, \boldsymbol{R}_s \epsilon_\theta(\boldsymbol{u}(s), s) \tag{52}$$

for the mean of $\hat{p}_{st}(\boldsymbol{u}(t)|\boldsymbol{u}(s))$. Its covariance $\boldsymbol{P}_{s\tau}$ satisfies

$$\frac{d\boldsymbol{P}_{s\tau}}{d\tau} = \hat{\boldsymbol{F}}_\tau \boldsymbol{P}_{s\tau} + \boldsymbol{P}_{s\tau} \hat{\boldsymbol{F}}_\tau^T + \lambda^2 \boldsymbol{G}_\tau \boldsymbol{G}_\tau^T, \quad \boldsymbol{P}_{ss} = 0. \tag{53}$$

Eq. (52) has a closed form expression with the help of the following lemma.

**Lemma 1.**

$$\int_s^t \hat{\Psi}(t,\tau) \frac{1+\lambda^2}{2} \boldsymbol{G}_\tau \boldsymbol{G}_\tau^T \Sigma_\tau^{-1} \Psi(\tau,s) = \hat{\Psi}(t,s) - \Psi(t,s)$$

*Proof.* For a fixed $s$, we define $\boldsymbol{N}(t) = \int_s^t \hat{\Psi}(t,\tau) \frac{1+\lambda^2}{2} \boldsymbol{G}_\tau \boldsymbol{G}_\tau^T \Sigma_\tau^{-1} \Psi(\tau,s)$ and $\boldsymbol{M}(t) = \hat{\Psi}(t,s) - \Psi(t,s)$. It follows that

$$\begin{aligned}
\frac{d\boldsymbol{N}(\tau)}{d\tau} &= \hat{\boldsymbol{F}}_\tau \boldsymbol{N}(\tau) + \frac{1+\lambda^2}{2} \boldsymbol{G}_\tau \boldsymbol{G}_\tau^T \Sigma_\tau^{-1} \Psi(\tau,s) \\
&= \boldsymbol{F}_\tau \boldsymbol{N}(\tau) + \frac{1+\lambda^2}{2} \boldsymbol{G}_\tau \boldsymbol{G}_\tau^T \Sigma_\tau^{-1} [\boldsymbol{N}(\tau) + \Psi(\tau,s)]
\end{aligned} \tag{54}$$

$$\begin{aligned}
\frac{d\boldsymbol{M}(\tau)}{d\tau} &= \hat{\boldsymbol{F}}_\tau \hat{\Psi}(\tau,s) - \boldsymbol{F}_\tau \Psi(\tau,s) \\
&= \boldsymbol{F}_\tau \boldsymbol{M}(\tau) + \frac{1+\lambda^2}{2} \boldsymbol{G}_\tau \boldsymbol{G}_\tau^T \Sigma_\tau^{-1} \hat{\Psi}(\tau,s).
\end{aligned} \tag{55}$$

Define $\boldsymbol{E}(t) = \boldsymbol{N}(t) - \boldsymbol{M}(t)$, then

$$\frac{d\boldsymbol{E}(t)}{dt} = (\boldsymbol{F}_t + \frac{1+\lambda^2}{2} \boldsymbol{G}_t \boldsymbol{G}_t^T \Sigma_t^{-1}) \boldsymbol{E}(t). \tag{56}$$

On the other hand, $\boldsymbol{N}(s) = \boldsymbol{M}(s) = 0$ which implies $\boldsymbol{E}(s) = 0$. We thus conclude $\boldsymbol{E}(t) = 0$ and $\boldsymbol{N}(t) = \boldsymbol{M}(t)$. $\qquad \square$

Using lemma 1, we simplify Eq. (52) to

$$\Psi(t,s)\boldsymbol{u}(s) + [\hat{\Psi}(t,s) - \Psi(t,s)]\boldsymbol{R}_s \epsilon_\theta(\boldsymbol{u}(s),s), \tag{57}$$

which is the mean in Eq. (22).

### B.3.3 PROOF OF THM 1

We restate the conclusion presented in Thm 1. The exact solution $\boldsymbol{u}(t - \Delta t)$ to Eq. (6) with coefficient Eq. (8) is

$$\boldsymbol{u}(t - \Delta t) \sim \mathcal{N}(\sqrt{\frac{\alpha_{t-\Delta t}}{\alpha_t}} \boldsymbol{u}(t) + \left[ -\sqrt{\frac{\alpha_{t-\Delta t}}{\alpha_t}} \sqrt{1 - \alpha_t} + \sqrt{1 - \alpha_{t-\Delta t} - \sigma_t^2} \right] \epsilon_\theta(\boldsymbol{u}(t), t), \sigma_t^2 \boldsymbol{I}_d) \tag{58}$$

with $\sigma_t^2 = (1 - \alpha_{t-\Delta t}) \left[ 1 - \left( \frac{1 - \alpha_{t-\Delta t}}{1 - \alpha_t} \right)^{\lambda^2} \left( \frac{\alpha_t}{\alpha_{t-\Delta t}} \right)^{\lambda^2} \right]$, which is the same as the DDIM Eq. (9).

Thm 1 is a concrete application of Eq. (22) when the DM is a DDPM and $\boldsymbol{F}_\tau, \boldsymbol{G}_\tau$ are set to Eq. (8). Thanks to the special form of $\boldsymbol{F}_\tau$, $\Psi$ has the expression

$$\Psi(t,s) = \sqrt{\frac{\alpha_t}{\alpha_s}} \boldsymbol{I}_d, \tag{59}$$

and $\hat{\Psi}$ satisfies

$$\log \hat{\Psi}(t,s) = \int_s^t \left[ \frac{1}{2} \frac{d \log \alpha_\tau}{d\tau} - \frac{1+\lambda^2}{2} \frac{d \log \alpha_\tau}{d\tau} \frac{1}{1 - \alpha_\tau} \right] d\tau \tag{60}$$

$$\hat{\Psi}(t,s) = \left( \frac{1 - \alpha_t}{1 - \alpha_s} \right)^{\frac{1+\lambda^2}{2}} \left( \frac{\alpha_s}{\alpha_t} \right)^{\frac{\lambda^2}{2}}. \tag{61}$$

**Mean:** Based on Eq. (22), we obtain the mean of $\hat{p}_{st}$ as

$$\sqrt{\frac{\alpha_t}{\alpha_s}}\boldsymbol{u}(s) + \left[-\sqrt{\frac{\alpha_t}{\alpha_s}}\sqrt{1-\alpha_s} + \left(\frac{1-\alpha_t}{1-\alpha_s}\right)^{\frac{1+\lambda^2}{2}} \left(\frac{\alpha_s}{\alpha_t}\right)^{\frac{\lambda^2}{2}}\sqrt{1-\alpha_s}\right]\epsilon_\theta(\boldsymbol{u}(s),s) \tag{62}$$

$$= \sqrt{\frac{\alpha_t}{\alpha_s}}\boldsymbol{u}(s) + \left[-\sqrt{\frac{\alpha_t}{\alpha_s}}\sqrt{1-\alpha_s} + \sqrt{(1-\alpha_t)\left(\frac{1-\alpha_{t-\Delta t}}{1-\alpha_t}\right)^{\lambda^2}\left(\frac{\alpha_t}{\alpha_{t-\Delta t}}\right)^{\lambda^2}}\right]\epsilon_\theta(\boldsymbol{u}(s),s) \tag{63}$$

$$= \sqrt{\frac{\alpha_t}{\alpha_s}}\boldsymbol{u}(s) + \left[-\sqrt{\frac{\alpha_t}{\alpha_s}}\sqrt{1-\alpha_s} + \sqrt{1-\alpha_t-\sigma_s^2}\right]\epsilon_\theta(\boldsymbol{u}(s),s), \tag{64}$$

where

$$\sigma_s^2 = (1-\alpha_t)\left[1 - \left(\frac{1-\alpha_t}{1-\alpha_s}\right)^{\lambda^2}\left(\frac{\alpha_s}{\alpha_t}\right)^{\lambda^2}\right]. \tag{65}$$

Setting $(s,t) \leftarrow (t, t-\Delta t)$, we arrive at the mean update in Eq. (14).

**Covariance:** It follows from

$$\frac{d\boldsymbol{P}_{s\tau}}{d\tau} = 2\left[\frac{d\log\alpha_\tau}{2d\tau} - \frac{1+\lambda^2}{2}\frac{d\log\alpha_\tau}{d\tau}\frac{1}{1-\alpha_\tau}\right]\boldsymbol{P}_{s\tau} - \lambda^2\frac{d\log\alpha_\tau}{d\tau}\boldsymbol{I}_d, \ \boldsymbol{P}_{ss} = 0$$

that

$$\boldsymbol{P}_{st} = (1-\alpha_t)\left[1 - \left(\frac{1-\alpha_t}{1-\alpha_s}\right)^{\lambda^2}\left(\frac{\alpha_s}{\alpha_t}\right)^{\lambda^2}\right].$$

Setting $(s,t) \leftarrow (t, t-\Delta t)$, we recover the covariance in Eq. (14).

### B.3.4 PROOF OF PROP 7

When $\lambda = 0$, the update step in Eq. (22) from $s$ to $t$ reads

$$\boldsymbol{u}(t) = \Psi(t,s)\boldsymbol{u}(s) + [\hat{\Psi}(t,s) - \Psi(t,s)]\boldsymbol{R}_s\epsilon_\theta(\boldsymbol{u}(s),s). \tag{66}$$

Meanwhile, the update step in Eq. (18) from $s$ to $t$ is

$$\boldsymbol{u}(t) = \Psi(t,s)\boldsymbol{u}(s) + \left[\int_s^t \frac{1}{2}\Psi(t,\tau)\boldsymbol{G}_\tau\boldsymbol{G}_\tau^T\boldsymbol{R}_\tau^{-T}d\tau\right]\epsilon_\theta(\boldsymbol{u}(s),s). \tag{67}$$

Eqs. (66) and (67) are equivalent once we have the following lemma.

**Lemma 2.** *When $\lambda = 0$,*

$$\int_s^t \frac{1}{2}\Psi(t,\tau)\boldsymbol{G}_\tau\boldsymbol{G}_\tau^T\boldsymbol{R}_\tau^{-T}d\tau = [\hat{\Psi}(t,s) - \Psi(t,s)]\boldsymbol{R}_s.$$

*Proof.* We introduce two new functions

$$\boldsymbol{N}(t) := \int_s^t \frac{1}{2}\Psi(t,\tau)\boldsymbol{G}_\tau\boldsymbol{G}_\tau^T\boldsymbol{R}_\tau^{-T}d\tau \tag{68}$$

$$\boldsymbol{M}(t) := [\hat{\Psi}(t,s) - \Psi(t,s)]\boldsymbol{R}_s. \tag{69}$$

First, $\boldsymbol{N}(s) = \boldsymbol{M}(s) = 0$. Second, they satisfy

$$\frac{d\boldsymbol{N}(t)}{dt} = \boldsymbol{F}_t\boldsymbol{N}(t) + \frac{1}{2}\boldsymbol{G}_t\boldsymbol{G}_t^T\boldsymbol{R}_t^{-T} \tag{70}$$

$$\frac{d\boldsymbol{M}(t)}{dt} = [\hat{\boldsymbol{F}}_t\hat{\Psi}(t,s) - \boldsymbol{F}_t\Psi(t,s)]\boldsymbol{R}_s \tag{71}$$

$$= \boldsymbol{F}_t\boldsymbol{M}(t) + \frac{1}{2}\boldsymbol{G}_t\boldsymbol{G}_t^T\Sigma_t^{-1}\hat{\Psi}(t,s)\boldsymbol{R}_s. \tag{72}$$

Note $\hat{\Psi}$ and $\boldsymbol{R}$ satisfy the same linear differential equation as

$$\frac{d\hat{\Psi}(t,s)}{dt} = [\boldsymbol{F}_t + \frac{1}{2}\boldsymbol{G}_t\boldsymbol{G}_t^T\Sigma_t^{-1}]\hat{\Psi}(t,s), \quad \frac{d\boldsymbol{R}_t}{dt} = [\boldsymbol{F}_t + \frac{1}{2}\boldsymbol{G}_t\boldsymbol{G}_t^T\Sigma_t^{-1}]\boldsymbol{R}_t. \tag{73}$$

It is a standard result in linear system theory (see Särkkä & Solin (2019, Eq(2.34))) that $\hat{\Psi}(t,s) = \boldsymbol{R}_t\boldsymbol{R}_s^{-1}$. Plugging it and $\boldsymbol{R}_t\boldsymbol{R}_t^T = \Sigma_t$ into Eq. (72) yields

$$\frac{d\boldsymbol{M}(t)}{dt} = \boldsymbol{F}_t\boldsymbol{M}(t) + \frac{1}{2}\boldsymbol{G}_t\boldsymbol{G}_t^T\boldsymbol{R}_t^{-T}. \tag{74}$$

Define $\boldsymbol{E}(t) = \boldsymbol{N}(t) - \boldsymbol{M}(t)$, then it satisfies

$$\boldsymbol{E}(s) = 0 \quad \frac{d\boldsymbol{E}(t)}{dt} = \boldsymbol{F}_t\boldsymbol{E}(t), \tag{75}$$

which clearly implies that $\boldsymbol{E}(t) = 0$. Thus, $\boldsymbol{N}(t) = \boldsymbol{M}(t)$. □

## C   MORE EXPERIMENT DETAILS

We present the practical implementation of gDDIM and its application to BMD and CLD. We include training details and discuss the necessary calculation overhead for executing gDDIM. More experiments are conducted to verify the effectiveness of gDDIM compared with other sampling algorithms. We report image sampling performance over an average of 3 runs with different random seeds.

### C.1   BDM: TRAINING AND SAMPLING

Unfortunately, the pre-trained models for BDM are not available. We reproduce the training pipeline in BDM (Hoogeboom & Salimans, 2022) to validate the acceleration of gDDIM. The official pipeline is quite similar to the popular DDPM (Ho et al., 2020). We highlight the main difference and changes in our implementation. Compared DDPM, BDM use a different forward noising scheme Eqs. (11) and (25). The two key hyperparameters $\{\boldsymbol{\alpha}_t\}, \{\boldsymbol{\sigma}_t\}$ follow the exact same setup in Hoogeboom & Salimans (2022), whose details and python implementation can be found in Appendix A (Hoogeboom & Salimans, 2022). In our implementation, we use Unet network architectures (Song et al., 2020b). We find our larger Unet improves samples quality. As a comparison, our SDE sampler can achieve FID as low as 2.51 while Hoogeboom & Salimans (2022) only has 3.17 on CIFAR10.

### C.2   CLD: TRAINING AND SAMPLING

For CLD, Our training pipeline, model architectures and hyperparameters are similar to those in Dockhorn et al. (2021). The main differences are in the choice of $\boldsymbol{K}_t$ and loss weights $\boldsymbol{K}_t^{-1}\Lambda_t\boldsymbol{K}_t^{-T}$.

Denote by $\epsilon_\theta(\boldsymbol{u},t) = [\epsilon_\theta(\boldsymbol{u},t;\boldsymbol{x}), \epsilon_\theta(\boldsymbol{u},t;\boldsymbol{v})]$ for corresponding model parameterization. The authors of Dockhorn et al. (2021) originally propose the parameterization $\boldsymbol{s}_\theta(\boldsymbol{u},t) = -\boldsymbol{L}_t^{-T}\epsilon_\theta(\boldsymbol{u},t)$ where $\Sigma_t = \boldsymbol{L}_t\boldsymbol{L}_t^T$ is the Cholesky decomposition of the covariance matrix of $p_{0t}(\boldsymbol{u}(t)|\boldsymbol{x}(0))$. Built on DSM Eq. (3), they propose *hybrid score matching (HSM)* that is claimed to be advantageous (Dockhorn et al., 2021). It uses the loss

$$\mathbb{E}_{t\sim\mathcal{U}[0,T]}\mathbb{E}_{\boldsymbol{x}(0),\boldsymbol{u}(t)|\boldsymbol{x}(0)}[\|\epsilon - \epsilon_\theta(\mu_t(\boldsymbol{x}_0) + \boldsymbol{L}_t\epsilon, t)\|_{\boldsymbol{L}_t^{-1}\Lambda_t\boldsymbol{L}_t^{-T}}^2]. \tag{76}$$

With a similar derivation (Dockhorn et al., 2021), we obtain the HSM loss with our new score parameterization $\boldsymbol{s}_\theta(\boldsymbol{u},t) = -\boldsymbol{L}_t^{-T}\epsilon_\theta(\boldsymbol{u},t)$ as

$$\mathbb{E}_{t\sim\mathcal{U}[0,T]}\mathbb{E}_{\boldsymbol{x}(0),\boldsymbol{u}(t)|\boldsymbol{x}(0)}[\|\epsilon - \epsilon_\theta(\mu_t(\boldsymbol{x}_0) + \boldsymbol{R}_t\epsilon, t)\|_{\boldsymbol{R}_t^{-1}\Lambda_t\boldsymbol{R}_t^{-T}}^2]. \tag{77}$$

Though Eqs. (76) and (77) look similar, we cannot directly use pretrained model provided in Dockhorn et al. (2021) for gDDIM. Due to the lower triangular structure of $\boldsymbol{L}_t$ and the special $\boldsymbol{G}_t$, the solution to Eq. (6) only relies on $\epsilon_\theta(\boldsymbol{u},t;\boldsymbol{v})$ and thus only $\epsilon_\theta(\boldsymbol{u},t;\boldsymbol{v})$ is learned in Dockhorn

et al. (2021) via a special choice of $\Lambda_t$. In contrast, in our new parametrization, both $\epsilon_\theta(\boldsymbol{u}, t; \boldsymbol{x})$ and $\epsilon_\theta(\boldsymbol{u}, t; \boldsymbol{v})$ are needed to solve Eq. (6). To train the score model for gDDIM, we set $\boldsymbol{R}_t^{-1} \Lambda_t \boldsymbol{R}_t^{-T} = \boldsymbol{I}$ for simplicity, similar to the choice made in Ho et al. (2020). Our weight choice has reasonable performance and we leave improvement possibilities, such as mixed score (Dockhorn et al., 2021), better $\Lambda_t$ weights (Song et al., 2021), for future work. Though we require a different training scheme of score model compared with Dockhorn et al. (2021), the modifications to the training pipeline and extra costs are almost ignorable.

We change from $\boldsymbol{K}_t = \boldsymbol{L}_t$ to $\boldsymbol{K}_t = \boldsymbol{R}_t$. Unlike $\boldsymbol{L}_t$ which has a triangular structure and closed form expression as (Dockhorn et al., 2021)

$$\boldsymbol{L}_t = \begin{bmatrix} \sqrt{\Sigma_t^{xx}} & 0 \\ \frac{\Sigma_t^{xv}}{\sqrt{\Sigma_t^{xx}}} & \sqrt{\frac{\Sigma_t^{xx}\Sigma_t^{xv} - (\Sigma_t^{xv})^2}{\Sigma_t^{xx}}} \end{bmatrix}, \quad \text{with} \quad \Sigma_t = \begin{bmatrix} \Sigma_t^{xx} & \Sigma_t^{xv} \\ \Sigma_t^{xv} & \Sigma_t^{vv} \end{bmatrix}, \tag{78}$$

we rely on high accurate numerical solver to solve $\boldsymbol{R}_t$. The triangular structure of $\boldsymbol{L}_t$ and sparse pattern of $\boldsymbol{G}_t$ for CLD in Eq. (10) also have an impact on the training loss function of the score model. Due to the special structure of $\boldsymbol{G}_t$, we only need to learn $\boldsymbol{s}_\theta(\boldsymbol{u}, t; \boldsymbol{v})$ signals present in the velocity channel. When $\boldsymbol{K}_t = \boldsymbol{L}_t$, $\boldsymbol{s}_\theta(\boldsymbol{u}, t) = -\boldsymbol{L}_t^{-T} \epsilon_\theta^{(\boldsymbol{L}_t)}(\boldsymbol{u}, t)$ with $\boldsymbol{L}_t^{-T}$ being upper triangular. Thus we only need to train $\epsilon_\theta^{(\boldsymbol{L}_t)}(\boldsymbol{u}, t; \boldsymbol{v})$ to recover $\boldsymbol{s}_\theta(\boldsymbol{u}, t; \boldsymbol{v})$. In contrast, $\boldsymbol{R}_t$ does not share the triangular structure as $\boldsymbol{L}_t$; both $\epsilon_\theta^{(\boldsymbol{R}_t)}(\boldsymbol{u}, t; \boldsymbol{x}), \epsilon_\theta^{(\boldsymbol{R}_t)}(\boldsymbol{u}, t; \boldsymbol{v})$ are needed to recover $\boldsymbol{s}_\theta(\boldsymbol{u}, t; \boldsymbol{v})$. Therefore, Dockhorn et al. (2021) sets loss weights

$$\boldsymbol{L}_t^{-1} \Lambda_t \boldsymbol{L}_t^{-T} = \begin{bmatrix} 0 & 0 \\ 0 & 1 \end{bmatrix} \otimes \boldsymbol{I}_d, \tag{79}$$

while we choose

$$\boldsymbol{R}_t^{-1} \Lambda_t \boldsymbol{R}_t^{-T} = \begin{bmatrix} 1 & 0 \\ 0 & 1 \end{bmatrix} \otimes \boldsymbol{I}_d. \tag{80}$$

As a result, we need to double channels in the output layer in our new parameterization associated with $\boldsymbol{R}_t$, though the increased number of parameters in last layer is negligible compared with other parts of diffusion models.

We include the model architectures and hyperparameters in Tab. 4. In additional to the standard size model on CIFAR10, we also train a smaller model for CELEBA to show the efficacy and advantages of gDDIM.

Table 4: Model architectures and hyperparameters

| Hyperparameter | CIFAR10 | CELEBA |
|---|---|---|
| Model | | |
| EMA rate | 0.9999 | 0.999 |
| # of ResBlock per resolution | 8 | 2 |
| Normalization | Group Normalization | Group Normalization |
| Progressive input | Residual | None |
| Progressive combine | Sum | N/A |
| Finite Impulse Response | Enabled | Disabled |
| Embedding type | Fourier | Positional |
| # of parameters | $\approx 108M$ | $\approx 62M$ |
| Training | | |
| # of iterations | 1m | 150k |
| Optimizer | Adam | Adam |
| Learning rate | $2 \times 10^{-4}$ | $2 \times 10^{-4}$ |
| Gradient norm clipping | 1.0 | 1.0 |
| Dropout | 0.1 | 0.1 |
| Batch size per GPU | 32 | 32 |
| GPUs | 4 A6000 | 4 A6000 |
| Training time | $\approx 79h$ | $\approx 16h$ |

### C.3 CALCULATION OF CONSTANT COEFFICIENTS

In gDDIM, many coefficients cannot be obtained in closed-form. Here we present our approach to obtain them numerically. Those constant coefficients can be divided into two categories, solutions to ODEs and definite integrals. We remark that these coefficients only need to be calculated once and then can be used everywhere. For CLD, each of these coefficient corresponds to a $2 \times 2$ matrix. The calculation of all these coefficients can be done within 1 min.

**Type I: Solving ODEs** The problem appears when we need to evaluate $\boldsymbol{R}_t$ in Eq. (17) and $\hat{\Psi}(t, s)$ in

$$\frac{d\hat{\Psi}(t,s)}{dt} = \hat{\boldsymbol{F}}_t \hat{\Psi}(t,s), \quad \hat{\Psi}(s,s) = \boldsymbol{I}. \tag{81}$$

Across our experiments, we use RK4 with a step size $10^{-6}$ to calculate the ODE solutions. For $\hat{\Psi}(t, s)$, we only need to calculate $\hat{\Psi}(t, 0)$ because $\hat{\Psi}(t, s) = \hat{\Psi}(t, 0)[\hat{\Psi}(s, 0)]^{-1}$. In CLD, $\boldsymbol{F}_t, \boldsymbol{G}_t, \boldsymbol{R}_t, \Sigma_t$ can be simplified to a $2 \times 2$ matrix; solving the ODE with a small step size is extremely fast. We note $\Psi(t, s)$ and $\Sigma_t$ admit close-form formula (Dockhorn et al., 2021). Since the output of numerical solvers are discrete in time, we employ a linear interpolation to handle query in continuous time. Since $\boldsymbol{R}_t, \hat{\Psi}$ are determined by the forward SDE in DMs, the numerical results can be shared. In stochastic gDDIM Eq. (22), we apply the same techniques to solve $\boldsymbol{P}_{st}$. In BDM, $\boldsymbol{F}_t$ can be simplified into matrix whose shape align with spatial shape of given images. We note that the drift coefficient of BDM is a diagonal matrix, we can decompose matrix ODE into multiple one dimensional ODE. Thanks to parallel computation in GPU, solving multiple one dimensional ODE is efficient.

**Type II: Definite integrals**

The problem appears in the derivation of update step in Eqs. (18), (19b) and (46), which require coefficients such as ${}^p\mathrm{C}_{ij}^{(q)}, {}^c\mathrm{C}_{ij}^{(q)}$. We use step size $10^{-5}$ for the integration from $t$ to $t - \Delta t$. The integrand can be efficiently evaluated in parallel using GPUs. Again, the coefficients, such as ${}^p\mathrm{C}_{ij}^{(q)}, {}^c\mathrm{C}_{ij}^{(q)}$, are calculated once and used afterwards if we need sample another batch with the same time discretization.

### C.4 GDDIM WITH OTHER DIFFUSION MODELS

Though we only test gDDIM in several existing diffusion models, including DDPM, BDM, and CLD, gDDIM can be applied to any other pre-trained diffusion models as long as the full score function is available. In the following, we list key procedures to integrate gDDIM sampler into general diffusion models. The integration consists of two stages, offline preparation of gDDIM (**Stage I**), and online execution of gDDIM (**Stage II**).

**Stage I: Offline preparation of gDDIM**  Preparation of gDDIM includes scheduling timestamps and the calculation of coefficients for the execution of gDDIM.

Step 1: Determine an increasing time sequence $\mathcal{T} = \{t_i\}$

Step 2: Obtain $\Psi(t, s)$ by solving ODE Eq. (81).

Step 3: Calculate $\boldsymbol{R}_t$ by solving ODE Eq. (17)

Step 4: Obtain ${}^p\mathrm{C}_{ij}^{(q)}, {}^c\mathrm{C}_{ij}^{(q)}$ via applying definite integrator solvers on Eqs. (18), (19b) and (46)

How to solve ODEs Eqs. (17) and (81) and definite integrals Eqs. (18), (19b) and (46) has been discussed in App. C.3.

**Stage II: Online execution of gDDIM**  Stage II employs high order EI-based ODE solvers for Eq. (82) with $\boldsymbol{K}_t = \boldsymbol{R}_t$. We include pseudo-code for simulating EI-multistep solvers in Algo 1. It mainly uses updates Eq. (40) and Eq. (45).

### C.5 More experiments on the choice of score parameterization

Here we present more experiments details and more experiments regarding Prop 1 and differences between the two parameterizations involving $\boldsymbol{R}_t$ and $\boldsymbol{L}_t$.

**Toy experiments:**

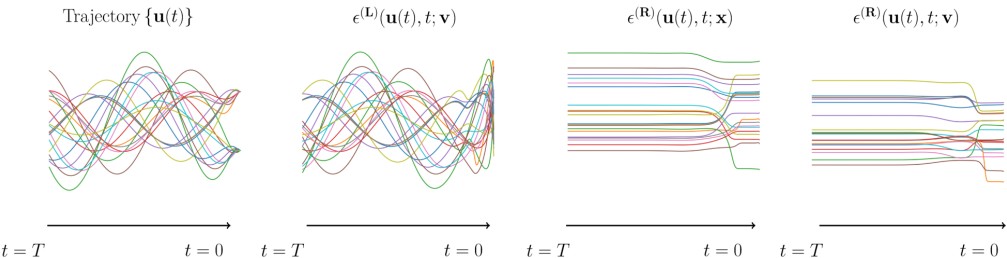

Figure 3: Trajectory and $\epsilon$ of Probability Flow solution in CLD

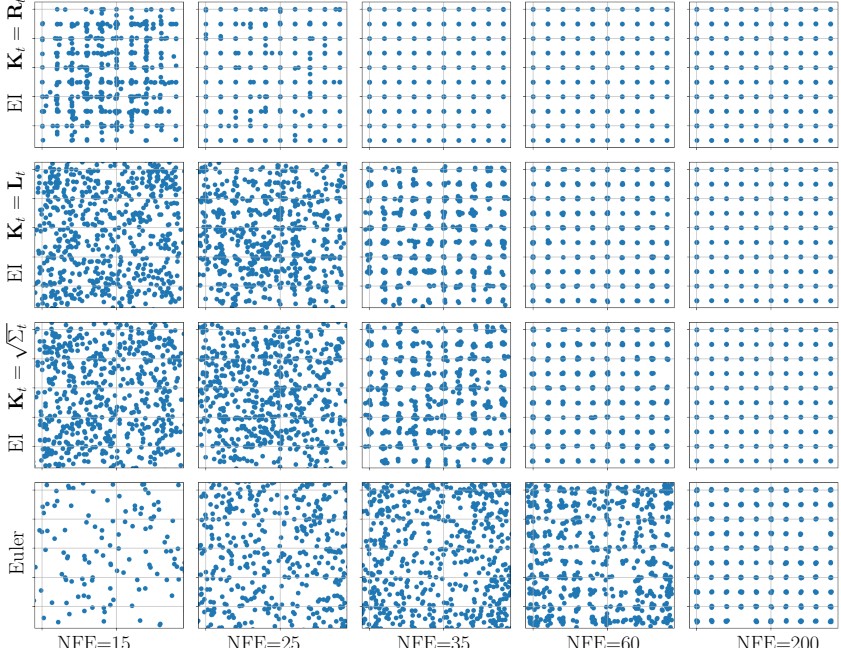

Figure 4: Sampling on a challenging 2D example with the exact score $\nabla \log p_t(\boldsymbol{u})$, where data distribution is a mixture of Gaussian with small variance. Compared with Euler, algorithms based on Exponential Integrator (EI) Eq. (82) have much better sampling quality. Among EI-based samplers, different $\boldsymbol{K}_t$ for score parameterization $\epsilon(\boldsymbol{u}, t) = -\boldsymbol{K}_t^{-T} \log p_t(\boldsymbol{u})$ have different sampling performances when NFE is small. Clearly, $\boldsymbol{R}_t$ proposed by gDDIM enjoys better sampling quality given the same NFE budget.

Here we present more empirical results to demonstrate the advantage of proper $\boldsymbol{K}_t$.

In VPSDE, the advantage of DDIM has been verified in various models and datasets (Song et al., 2020a). To empirically verify Prop 1, we present one toy example where the data distribution is a mixture of two one dimension Gaussian distributions. The ground truth $\nabla \log p_t(\boldsymbol{u})$ is known in this toy example. As shown in Fig. 2, along the solution to probability flow ODE, the score parameterization $\epsilon(\boldsymbol{u}, t) = -\nabla \log p_t(\boldsymbol{u})\sqrt{1 - \alpha_t}$ enjoys smoothing property. We remark that $\boldsymbol{R}_t = \boldsymbol{L}_t = \sqrt{\Sigma_t} = \sqrt{1 - \alpha_t}\boldsymbol{I}_d$ in VPSDE, and thus gDDIM is the same as DDIM; the differences among $\boldsymbol{R}_t, \boldsymbol{L}_t, \sqrt{\Sigma_t}$ only appear when $\Sigma_t$ is non-diagonal.

| $q$ | $K_t$ | FID / IS at different NFE | | | |
|---|---|---|---|---|---|
| | | 20 | 30 | 40 | 50 |
| 0 | $L_t$ | 461.72 / 1.18 | 441.05 / 1.21 | 244.26 / 2.39 | 120.07 / 5.25 |
| | $R_t$ | 16.74 / 8.67 | 9.73 / 8.98 | 6.32 / 9.24 | 5.17 / 9.39 |
| 1 | $L_t$ | 368.38 / 1.32 | 232.39 / 2.43 | 99.45 / 4.92 | 57.06 / 6.70 |
| | $R_t$ | 8.03 / 9.12 | 4.26 / 9.49 | 3.27 / 9.61 | 2.67 / 9.65 |
| 2 | $L_t$ | 463.33 / 1.17 | 166.90 / 3.56 | 4.12 / 9.25 | 3.31 / 9.38 |
| | $R_t$ | **3.90 / 9.56** | **2.66 / 9.64** | **2.39 / 9.74** | 2.32 / 9.75 |
| 3 | $L_t$ | 464.36 / 1.17 | 463.45 / 1.17 | 463.32 / 1.17 | 240.45 / 2.29 |
| | $R_t$ | 332.70 / 1.47 | 292.31 / 1.70 | 13.27 / 10.15 | **2.26 / 9.77** |

Table 5: More experiments on CIFAR10

| $q$ | $K_t$ | FID at different NFE | | | |
|---|---|---|---|---|---|
| | | 20 | 30 | 40 | 50 |
| 0 | $L_t$ | 446.56 | 430.59 | 434.79 | 379.73 |
| | $R_t$ | 37.72 | 13.65 | 12.51 | 8.96 |
| 1 | $L_t$ | 261.90 | 123.49 | 105.75 | 88.31 |
| | $R_t$ | 12.68 | 7.78 | 5.93 | 5.11 |
| 2 | $L_t$ | 446.74 | 277.28 | 6.18 | 5.48 |
| | $R_t$ | **6.86** | **5.67** | **4.62** | 4.19 |
| 3 | $L_t$ | 449.21 | 440.84 | 443.91 | 286.22 |
| | $R_t$ | 386.14 | 349.48 | 20.14 | **3.85** |

Table 6: More experiments on CELEBA

In CLD, we present a similar study. As the covariance $\Sigma_t$ in CLD is no longer diagonal, we find the difference of the $L_t$ parameterization suggested by (Dockhorn et al., 2021) and the $R_t$ parameterization is large in Fig. 3. The oscillation of $\{\epsilon^{(L_t)}\}$ prevents numerical solvers from taking large step sizes and slows down sampling. We also include a more challenging 2D example to illustrate the difference further in Fig. 4. We compare sampling algorithms based on Exponential Integrator without multistep methods for a fair comparison, which reads

$$\boldsymbol{u}(t - \Delta t) = \Psi(t - \Delta t, t)\boldsymbol{u}(t) + [\int_t^{t - \Delta t} \frac{1}{2}\Psi(t - \Delta t, \tau)\boldsymbol{G}_\tau \boldsymbol{G}_\tau^T \boldsymbol{K}_\tau^{-T} d\tau]\epsilon^{(\boldsymbol{K}_t)}(\boldsymbol{u}, t), \quad (82)$$

where $\epsilon^{(\boldsymbol{K}_t)}(\boldsymbol{u}, t) = \boldsymbol{K}_t^T \nabla \log p_t(\boldsymbol{u}(t))$. Though we have the exact score, sampling with Eq. (82) will not give us satisfying results if we use $\boldsymbol{K}_t = \boldsymbol{L}_t$ other than $\boldsymbol{R}_t$ when NFE is small.

**Image experiments:**

We present more empirical results regarding the comparison between $\boldsymbol{L}_t$ and $\boldsymbol{R}_t$. Note that we use exponential integrator for the parametrization $\boldsymbol{L}_t$ as well, similar to DEIS (Zhang & Chen, 2022). We vary the polynomial order $q$ in multistep methods (Zhang & Chen, 2022) and test sampling performance on CIFAR10 and CELEBA. In both datasets, we generate $50k$ images and calcualte their FID. As shown in Tabs. 5 and 6, $\boldsymbol{R}_t$ has significant advantages, especially when NFE is small.

We also find that multistep method with a large $q$ can harm sampling performance when NFE is small. This is reasonable; the method with larger $q$ assumes the nonlinear function is smooth in a large domain and may rely on outdated information for approximation, which may worsen the accuracy.

## C.6 MORE EXPERIMENTS ON THE CHOICE OF $\lambda$

To study the effects of $\lambda$, we visualize the trajectories generated with various $\lambda$ but the same random seeds in Fig. 5 on our toy example. Clearly, trajectories with smaller $\lambda$ have better smoothing property while trajectories with large $\lambda$ contain much more randomness. From the fast sampling perspective, trajectories with more stochasticity are much harder to predict with small NFE compared with smooth trajectories.

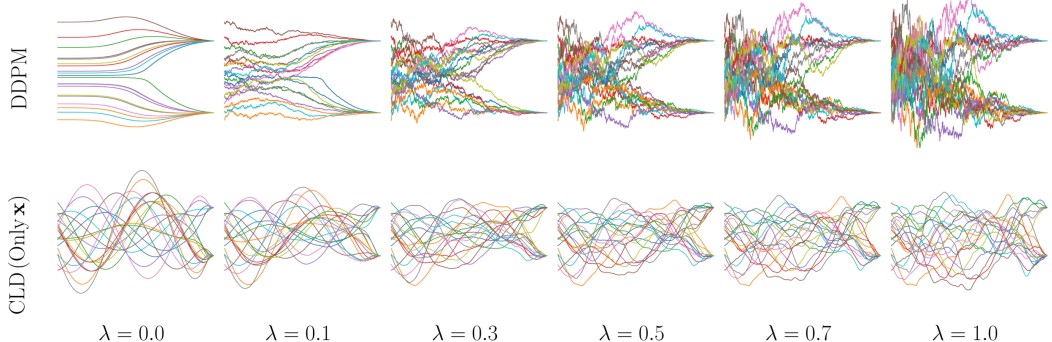

Figure 5: Sampling with various $\lambda$ and accurate score on a Toy example. Trajectories are shown from $T$ to 0.

We include more qualitative results on the choice of $\lambda$ and comparison between the Euler-Maruyama (EM) method and the gDDIM in Figs. 8 and 9. Clearly, when NFE is small, increasing $\lambda$ has a negative effect on the sampling quality of gDDIM. We hypothesize that $\lambda = 0$ already generates high-fidelity samples and additional noise may harm the sampling performance. With a fixed number of function evaluations, information derived from score network fails to remove the injected noise as we increase $\lambda$. On the other hand, we find that the EM method shows slightly better quality as we increase $\lambda$. We hypothesize that the ODE or SDEs with small $\lambda$ has more oscillations than SDEs with large $\lambda$. It is known that the EM method has a very bad performance for oscillations systems and suffers from large discretization error (Press et al., 2007). From previous experiments, we find that ODE in CLD is highly oscillated.

We also find both methods perform worse than Symmetric Splitting CLD Sampler (SSCS) (Dockhorn et al., 2021) when $\lambda = 1$. The improvement by utilizing Hamiltonian structure and SDEs structure is significant. This encourages further exploration that incorporates Hamiltonian structure into gDDIM in the future. Nevertheless, we also remark that SSCS with $\lambda = 1.0$ performs much worse than gDDIM with $\lambda = 0$.

## C.7    MORE COMPARISONS

We also compare the performance of the CLD model we trained with that claimed in Dockhorn et al. (2021) in Tab. 7. We find that our trained model performs worse than Dockhorn et al. (2021) when a blackbox ODE solver or EM sampling scheme with large NFE are used. There may be two reasons. First, with similar size model, our training scheme not only needs to fit $\nabla_{\boldsymbol{v}} \log p_t(\boldsymbol{u})$, but also $\nabla_{\boldsymbol{x}} \log p_t(\boldsymbol{u})$, while Dockhorn et al. (2021) can allocate all representation resources of neural network to $\nabla_{\boldsymbol{v}} \log p_t(\boldsymbol{u})$. Another factor is the mixed score trick on parameterization, which is shown empirically have a boost in model performance (Dockhorn et al., 2021) but we do not include it in our training.

We also compare our algorithm with more accelerating sampling methods in Tab. 7. gDDIM has achieved the best sampling acceleration results among training-free methods, but it still cannot compete with some distillation-based acceleration methods. In Tab. 8, we compare Predictor-only method with Predictor-Corrector (PC) method. With the same number of steps $N$, PC can improve the quality of Predictor-only at the cost of additional $N - 1$ score evaluations, which is almost two times slower compared with the Predictor-only method. We also find large $q$ may harm the sampling performance in the exponential multistep method when NFE is small. We note high order polynomial requires more datapoints to fit polynomial. The used datapoints may be out-of-date and harmful to sampling quality when we have large stepsizes.

## C.8    NEGATIVE LOG LIKELIHOOD EVALUATION

Because our method only modifies the score parameterization compared with the original CLD (Dockhorn et al., 2021), we follow a similar procedure to evaluate the bound of negative log-likelihood (NLL). Specifically, we can simulate probability ODE Eq. (7) to estimate the log-

Table 7: More comparison on CIFAR10. FIDs may be reported based on different training techniques and data augmentation. It should not be regarded as the only evidence to compare different algorithms.

| Class | Model | NFE ($\downarrow$) | FID ($\downarrow$) |
|---|---|---|---|
| **CLD** | our CLD-SGM (gDDIM) | 50 | 2.26 |
| | our CLD-SGM (SDE, EM) | 2000 | 2.39 |
| | our CLD-SGM (Prob.Flow, RK45) | 155 | 2.86 |
| | our CLD-SGM (Prob.Flow, RK45) | 312 | 2.26 |
| | CLD-SGM (Prob.Flow, RK45) by Dockhorn et al. (2021) | 147 | 2.71 |
| | CLD-SGM (SDE, EM) by Dockhorn et al. (2021) | 2000 | 2.23 |
| **Training-free Accelerated Score** | DDIM by Song et al. (2020a) | 100 | 4.16 |
| | Gotta Go Fast by Jolicoeur-Martineau et al. (2021b) | 151 | 2.73 |
| | Analytic-DPM by Bao et al. (2022) | 100 | 3.55 |
| | FastDPM by Kong & Ping (2021a) | 100 | 2.86 |
| | PNDM by Liu et al. (2022) | 100 | 3.53 |
| | DEIS by Zhang & Chen (2022) | 50 | 2.56 |
| | DPM-Solver by Lu et al. (2022) | 50 | 2.65 |
| | Rescaled $2^{nd}$ Order Heun by Karras et al. (2022) | 35 | 1.97 |
| **Training-needed Accelerated Score** | DDSS by Watson et al. (2022) | 25 | 4.25 |
| | Progressive Distillation by Salimans & Ho (2022) | 4 | 3.0 |
| | Knowledge distillation by Luhman & Luhman (2021) | 1 | 9.36 |
| **Score+Others** | LSGM by Vahdat et al. (2021) | 138 | 2.10 |
| | LSGM-100M by Vahdat et al. (2021) | 131 | 4.60 |
| | Diffusion GAN by Xiao et al. (2021) | 4 | 3.75 |

| $q$ | Method | \multicolumn{4}{c}{FID at different steps $N$} | | | |
|---|---|---|---|---|---|
| | | 20 | 30 | 40 | 50 |
| 0 | Predictor | 16.74 | 9.73 | 6.32 | 5.17 |
| 1 | Predictor | 8.03 | 4.26 | 3.27 | 2.67 |
| | PC | 6.24 | 2.36 | 2.26 | 2.25 |
| 2 | Predictor | 3.90 | 2.66 | 2.39 | 2.32 |
| | PC | 3.01 | 2.29 | 2.26 | 2.26 |
| 3 | Predictor | 332.70 | 292.31 | 13.27 | 2.26 |
| | PC | 337.20 | 313.21 | 2.67 | 2.25 |

Table 8: Predictor-only vs Predictor-Corrector (PC). Compared with Predictor-only, PC adds one more correcting step after each predicting step except the last step. When sampling with $N$ step, the Predictor-only approach requires $n$ score evaluation, while PC consumes $2N - 1$ NFE.

likelihood of given data (Grathwohl et al., 2018). However, our diffusion model models the joint distribution $p(\boldsymbol{u}_0) = p(\boldsymbol{x}_0, \boldsymbol{v}_0)$ on test data $\boldsymbol{x}_0$ and augmented velocity data $\boldsymbol{v}_0$. Getting marginal distribution $p(\boldsymbol{x}_0)$ from $p(\boldsymbol{u}_0)$ is challenging, as we need integrate $\boldsymbol{v}_0$ for each $\boldsymbol{x}_0$. To circumvent this issue, Dockhorn et al. (2021) derives a lower bound on the log-likelihood,

$$\log p(\boldsymbol{x}_0) = \log(\int p(\boldsymbol{v}_0)\frac{p(\boldsymbol{x}_0, \boldsymbol{v}_0)}{p(\boldsymbol{v}_0}) d\boldsymbol{v}_0)$$
$$\geq \mathbb{E}_{\boldsymbol{v}_0 \sim p(\boldsymbol{v}_0)}[\log p(\boldsymbol{x}_0, \boldsymbol{v}_0)] + H(p(\boldsymbol{v}_0)),$$

where $H(p(\boldsymbol{v}_0))$ denotes the entropy of $p(\boldsymbol{v}_0)$. We can then estimate the lower bound with the Monte Carlo approach.

Empirically, our trained model achieves a NLL upper bound 3.33 bits/dim, which is comparable with 3.31 bits/dim reported in the original CLD (Dockhorn et al., 2021). Possible approaches to further reduce the NLL bound include maximal likelihood weights (Song et al., 2021), and improved training techniques such as mixed score. For more discussions of log-likelihood and how to tighten the bound, we refer the reader to Dockhorn et al. (2021).

## C.9 CODE LICENSES

We implemented gDDIM and related algorithms in Jax. We have used code from a number of sources in Tab. 9.

| URL | Citation | License |
| --- | --- | --- |
| https://github.com/yang-song/score_sde | Song et al. (2020b) | Apache License 2.0 |
| https://github.com/nv-tlabs/CLD-SGM | Dockhorn et al. (2021) | NVIDIA License |
| https://github.com/qsh-zh/deis | Zhang & Chen (2022) | Unknown |

Table 9: Code License

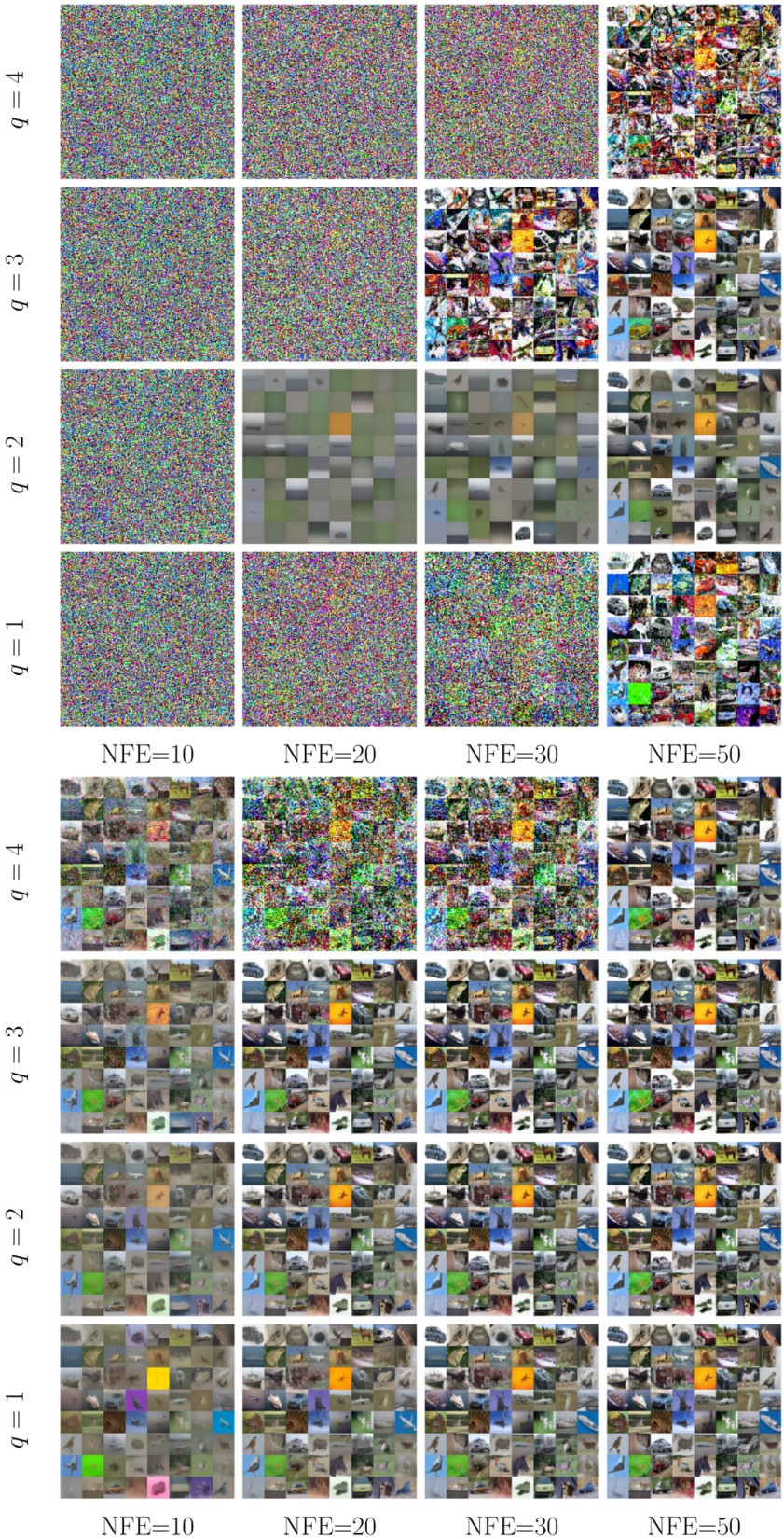

Figure 6: Comparison between $\boldsymbol{L}$ (Upper) and $\boldsymbol{R}$ (Lower) with exponential integrator on CIFAR10.

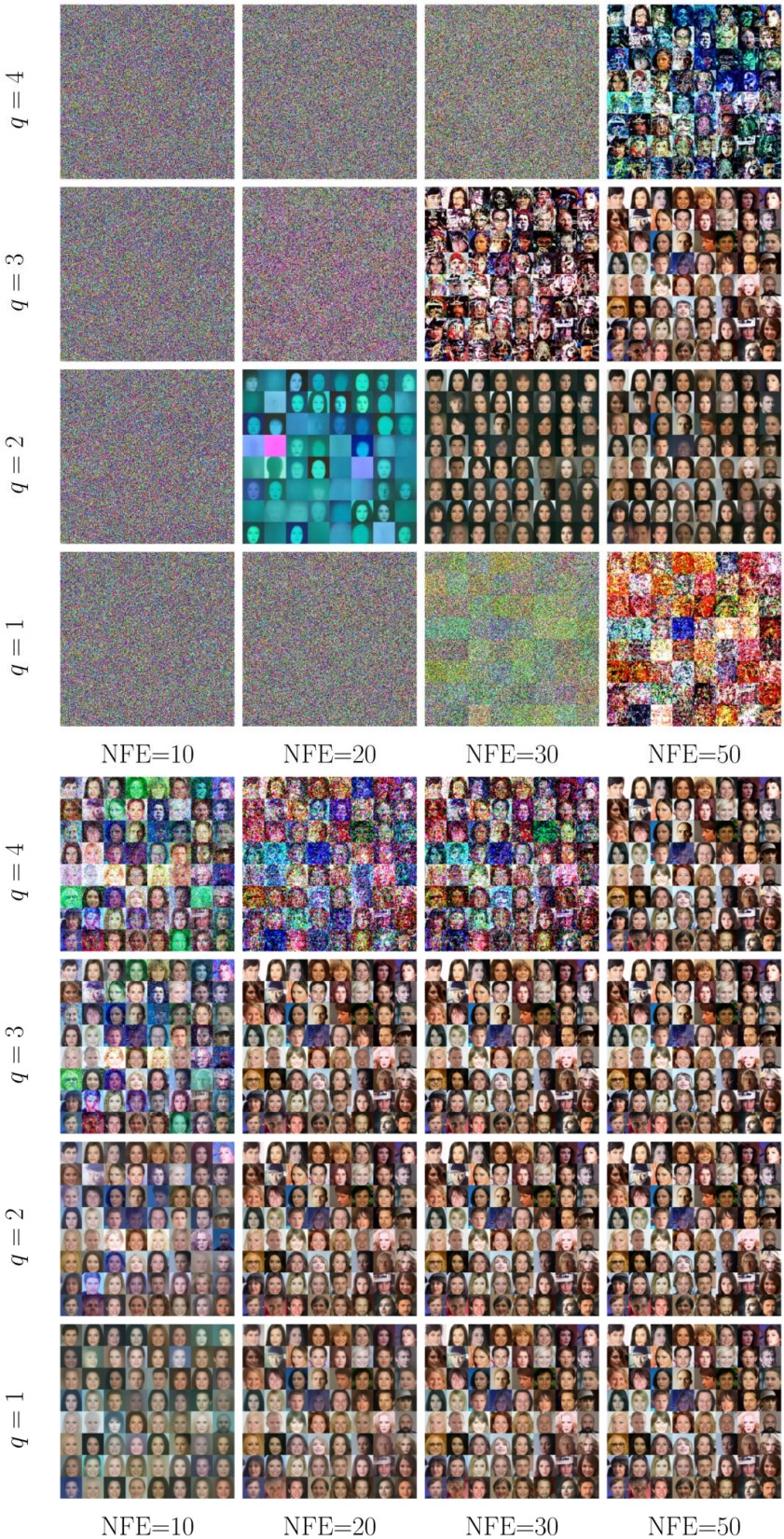

Figure 7: Comparison between $L$ (Upper) and $R$ (Lower) with exponential integrator on CELEBA.

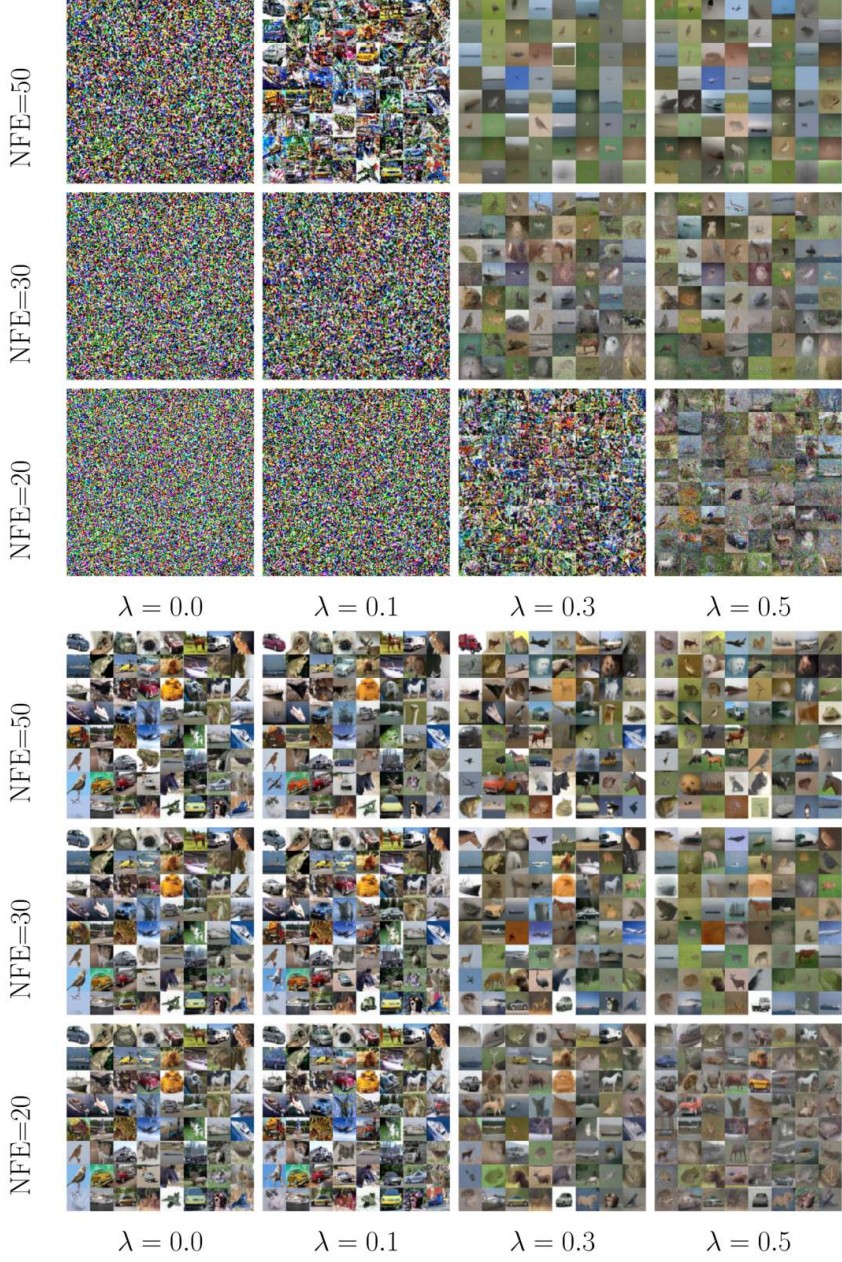

Figure 8: Comparison between EM (Upper) and gDDIM (Lower) on CIFAR10.

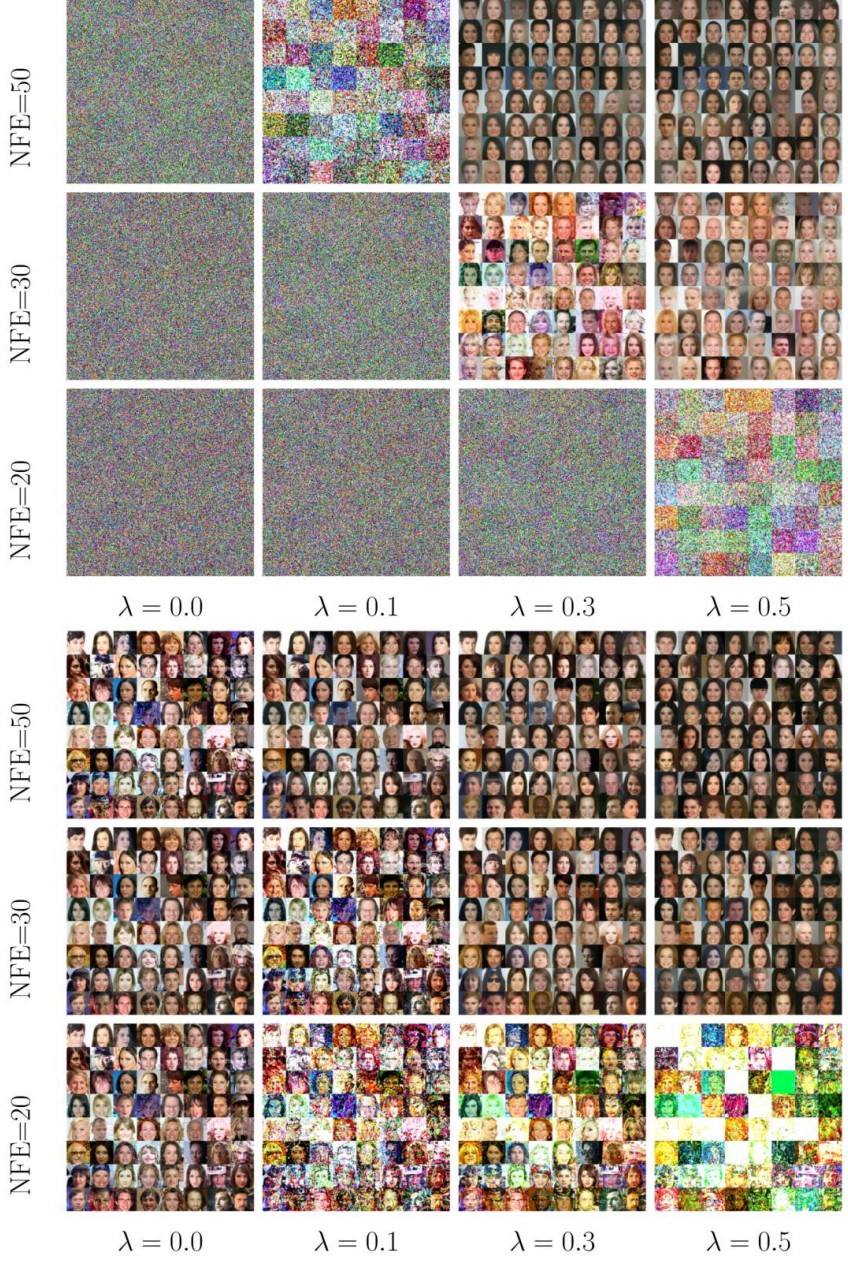

Figure 9: Comparison between EM (Upper) and gDDIM (Lower) on CELEBA.

