# OpenReview forum: "gDDIM: Generalized denoising diffusion implicit models"
_ICLR.cc/2023/Conference — ICLR 2023 notable top 25%_

### Official Review · Reviewer_85mY · 2022-10-25

**Confidence:** 3
**Correctness:** 3
**Technical Novelty And Significance:** 4
**Empirical Novelty And Significance:** 3
**Recommendation:** 8

**Clarity, Quality, Novelty And Reproducibility:**

Clarity:

Section 3: The authors state that the Euler and Runge Kutta solvers on the ODE cannot obtain a single-step recovery of a data point, while DDIM can. However, the original work on DDIMs [1] appear to contradict this claim: [1] states that DDIMs are probability flow ODEs of the variance exploding diffusion SDE as described in [2].

Proposition 3 states that Eq. 13 holds when the data is a Dirac distribution for a single data point. However, Thm 1 assumes that Eq. 13 is approximately true for general data. Why is this a valid assumption?

Dirac $p_0(u)$ and the manifold hypothesis: I found the reasoning in this section unclear. Can the authors elaborate on why the manifold hypothesis explains the effectiveness of DDIMs?

Does the integral in Eq. 18 have a closed form solution? If not, how is it computed during sampling?

[1]  Song, J., Meng, C. and Ermon, S., 2020. Denoising diffusion implicit models. arXiv preprint arXiv:2010.02502.

[2] Song, Y., Sohl-Dickstein, J., Kingma, D.P., Kumar, A., Ermon, S. and Poole, B., 2020. Score-based generative modeling through stochastic differential equations. arXiv preprint arXiv:2011.13456.

**Strength And Weaknesses:**

Strengths:
- The work proposes a generalized framework for deriving accelerated samplers from standard diffusion models.
- The proposed framework performs well empirically, and allows one to make informed choices on how the learned score function should be reparameterized (e.g. CLD appears to be much improved in the low NFE scheme by choosing $K_t = L_t$).

Weaknesses:
- While the derivation produces the original DDIM, when applied to DDPM models, it involves several steps that make questionable assumptions (e.g., Section 3 assumes unperturbed data is a Dirac distribution on a single point, Section 4 assumes data is normally distributed.)

**Summary Of The Paper:**

The authors propose a theoretical framework that extends DDIMs, which were derived only for DDPMs, to a more general form of diffusion SDEs.

**Summary Of The Review:**

Overall, the contribution of this work is simple yet significant. While the proposed framework for building generalized DDIMs relies on several somewhat unrealistic assumptions (see first point in Weaknesses), these choices are at least partially justified by the promising empirical performance of the resulting samplers.

---

> ### Author Response · Authors · 2022-11-15
> **Response**
>
> We appreciate the detailed reviews and thoughtful feedback from the reviewer.
>
> ********Q: Questionable assumptions. Section 3 assumes unperturbed data is a Dirac distribution on a single point, but Section 4 assumes data is normally distributed.********
>
> A: We start with the single data point assumption to explain the effectiveness and property of the popular DDIM. We derive the underlying principle used in DDIM from this simplest case and extend them to cases with real datasets, where the data distribution can be viewed as a mixture of Dirac (empirical data distribution).
>
> The Gaussian distribution is a generalization of Dirac distribution whose variance reduces to zero. The reason we have this generalization is that diffusion models with augmented states such as CLD use a Gaussian distribution on the velocity channel for each data point. Thus, when there is a single data point, the initial distribution of the forward process is a Gaussian instead of a Dirac distribution. We have added a sentence to clarify this point.
>
> ********Q: The authors state that the Euler and Runge Kutta solvers on the ODE cannot obtain a single-step recovery of a data point, while DDIM can. However, the original work on DDIMs [1] appear to contradict this claim: [1] states that DDIMs are probability flow ODEs of the variance exploding diffusion SDE as described in [2].********
>
> A: We are glad the reviewer raises this point. Directly applying the Euler or the Runge Kutta solver to ODE~(Eq 7 in our manuscript) does not give a single-step recovery of a data point due to discretization error, even when the data distribution is a Dirac. The reason why DDIM works is that it uses change-of-variable and handles semi-linear and $\mathbf{F}_t, \mathbf{G}_t$ analytically as in Eq-13/14 [1]. Though the authors of DDIM [1] did not point it out, the treatment is exactly the exponential integrator method we discussed in Proposition 2. In Sec 3 we provide an intuitive explanation of why DDIM works and can recover the data point in one-step when the data distribution is a Dirac. We would like to emphasize that DDIM is a numerical scheme for the probability flow ODE (7) and has discretization error in general for general data distributions.
>
> ********Q: However, Thm 1 assumes that Eq. 13 is approximately true for general data. Why is this a valid assumption?********
>
> A: The approximation is based on the observation that images in training datasets are far away from each other; an intuitive illustration is provided in Fig2. The approximation is exact if we have only one data point, and relatively accurate if the score of target distribution is close to the score of one dominant data point. To see this, consider Equation 15 and Fig2. In the red region in Fig 2, the noised data distribution $p_t(u) \approx p_{0t}(u(t)|u(0))$ and weights $\{w_m\}$ is dominated by one specific data in Eq 15. On the other hand, in the green region, the noised data is dominated by Gaussian noise and the Gaussian approximation is also reliable.
>
> In summary, the approximation is relatively accurate if the noised data distribution $p_t(u)$  is close to noised data distribution with respect to one point $p_{0t}(u(t)|u(0))$. The fact that realistic data is distributed sparsely in the data/pixel space implies this is the case in the red region.
>
> ****Q: I found the reasoning in Section 3 unclear. Can the authors elaborate on why the manifold hypothesis explains the effectiveness of DDIMs?****
>
> A: We first demonstrate that the approximation (Prop 1, 3) can recover DDIMs (Prop 2, 4). The approximations are first introduced via the simple scenario with one data point dataset before extending them into the general datasets.
>
> To see why DDIM is effective in general datasets, we notice that the score of general distribution can be approximated locally by the score of one data point dataset when t is small. This is because realistic datas are sparsely distributed in pixel space such that the distance between two images can be large even if they look similar. This is sometimes viewed as a consequence of the manifold hypothesis. As a consequence, the score distribution of realistic datasets consisted of well-separated data points may be dominated by one single datapoint as we illustrate in Fig 2 (red region). The observation is formally captured by Eq-15. This fact partially explains why DDIM and gDDIM can accelerate diffusion models on real datasets like images.

---

> > ### Author Response · Authors · 2022-11-15
> > **Response2**
> >
> > ********Q: Does the integral in Eq. 18 have a closed-form solution? If not, how is it computed during sampling?********
> >
> > A: If we have a simple diffusion model like DDPM, it has a closed-form solution as is the case in DDIM. For the non-isotropic models considered in this work, it is difficult to derive a closed-form solution. In practical implementation, we use a high accuracy numerical solver for calculation; the integral is relatively simple to compute. For instance, in CLD it is 2x2 matrix integration and any high accurate solver can be adopted to solve Eq. 18 efficiently. The computation is cheap in modern computers; it takes less than 1 second to calculate them and this computation only needs to be carried out once for a given time discretization. We also include a clean implementation of it in our attached codebase and more discussion in Section C.3.
> >
> > [1] J. Song et al. Denoising Diffusion Implicit Models
> >
> > [2] Y. Song et al. Score-based generative modeling through stochastic differential equations

---

### Official Review · Reviewer_vCLj · 2022-10-25

**Confidence:** 4
**Correctness:** 4
**Technical Novelty And Significance:** 4
**Empirical Novelty And Significance:** 4
**Recommendation:** 8

**Clarity, Quality, Novelty And Reproducibility:**

(+) This paper provides sound mathematical backups for gDDIM.

(-) The terms are excessively omitted. E.g., time-varying noise schedule $R_t$ -> time-varying $R_t$

(+) The theoretic results and applications are original.

**Strength And Weaknesses:**


Strengths

(+) The proposed generalized DDIM is versatile to different diffusion models.

(+) Step-by-step logical flow: Dirac distribution to Gaussian distribution to real distribution, deterministic case to stochastic case, etc.

(+) Straightforward illustration for understanding ideas.

Weaknesses

(-) Connection from Dirac distribution to the real distribution appears at the bottom of page 6. Bringing it to the top of Section 3 and describing "empirical distribution with weights vs Dirac = real distribution vs one sample" would provide the whole picture to the readers.
(-) Missing link: how do we move from Gaussian and to real distributions? I do not find assumptions or proofs.

Misc.
(-) Please leave vspace for readability, e.g., Figure 1.
(-) Please place figures close to its connected paragraphs, e.g., Figure 1.
(-) In Table 4, please fix the typo "# of arameters".


**Summary Of The Paper:**

This paper explains that the exact solution can be numerically found in DDIM with fast sampling.
Extending this explanation, this paper proposes "generalized DDIM" that modifies parameterization of the score networks of existing diffusion models to achieve 20x acceleration and comparable or much better FID even on non-isotropic diffusion models.

**Summary Of The Review:**

This paper shows non-trivial advance of diffusion models in identifying principles of DDIM and enabling fast sampling. Though its content is valuable, the presentation can be nicer to be easier to follow.

---

> ### Author Response · Authors · 2022-11-15
> **Response**
>
> We appreciate the detailed reviews and thoughtful feedback from the reviewer.
>
> ********Q: Connection from Dirac distribution to the real distribution appears at the bottom of page 6. Bringing it to the top of Section 3 and describing "empirical distribution with weights vs Dirac = real distribution vs one sample" would provide the whole picture to the readers.********
>
> A: We appreciate the suggestion. Section 3 starts with a simple example, which is followed by discussions on its generalization to general settings. We believe this way of presentation can prepare the reader before he/she gets to the more complicate general settings. Thus, we chose to keep the current organization of Section 3. However, we have added a few sentences at the beginning of Section 3 to provide the reader with an overall picture.
>
> ********Q: how do we move from Gaussian to real distributions? I do not find assumptions or proofs.********
>
> A: The connection from Gaussian to real distribution is a generalization of the connection from Dirac distribution to the real distributions. The reason we have this generalization is that diffusion models with augmented states such as CLD use a Gaussian distribution on the velocity channel for each data point. Thus, when there is a single data point, the initial distribution of the forward process is a Gaussian instead of a Dirac distribution. We have added a sentence to clarify this point.
>
> **************************************************************************Q: Improve paper readability, such as the usage of space.**************************************************************************
>
> A: We thank the reviewer for pointing those out. We have made modifications to our manuscript accordingly. We note that Figure 1 is an illustration of the main idea of the paper. We believe the current location of the figure may help the reader to have a good understanding of gDDIM quickly.

---

> > ### Comment · Reviewer_vCLj · 2022-11-19
> > **Thank you**
> >
> > Thank you for the reply.
> >
> > I hope to see this paper published in ICLR 2023.

---

### Official Review · Reviewer_PjHX · 2022-10-26

**Confidence:** 3
**Correctness:** 4
**Technical Novelty And Significance:** 1
**Empirical Novelty And Significance:** 3
**Recommendation:** 8

**Clarity, Quality, Novelty And Reproducibility:**

This is a well written paper with a clearly explained idea. The proposed gDDIM is novel and original, and the explanations for the success of DDIM are also novel.


**Strength And Weaknesses:**

Strength
- This is the first work to generalize DDIMs (gDDIM) using a theoretically grounded framework
- It provides insightful interpretations for the DDIMs that explains why ODE samplers perform well
- The reported experiments show strong improvements over the original samplers
- The paper is well written and contains sufficient details

Weakness
- The results could be more solid if tested with more realistic datasets beyond the toy CIFAR-10 such as for LSUN and ImageNet


**Summary Of The Paper:**

This submission deals with DDIMs. It proposes a generalization of DDIMs to general diffusion models beyond the isotropic denoising by modifying the parameterization of score. It links DDIMs to SDEs by a score approximation, and provides an interpretation for the success of DDIMs. It validates the results for BDM and CLD settings, where it achieves strong FID scores for CIFAR-10 sample generation under a small number of function evaluations compared with the original samplers.

**Summary Of The Review:**

This paper addresses an important and timely problem. It proposes a theoretically grounded gDDIM framework that generalized DDIM beyond isotropic denoising, and provides insightful interpretations for the good performance of DDIMs. The experiments also show significant improvement over original samplers.

---

> ### Author Response · Authors · 2022-11-15
> **Response**
>
> We appreciate the detailed reviews and encouragement from the reviewer.
>
> ********Q: The results could be more solid if tested with more realistic datasets beyond the toy CIFAR-10 such as for LSUN and ImageNet********
>
> A: gDDIM can be plugged into pre-trained score models without further training. One distinguishing feature of gDDIM is that it can be applied to non-isotropic diffusion models such as CLD and BDM. Unfortunately, existing works exploring various non-isotropic diffusion models are much more scarce compared with isotropic diffusion models, and pre-trained non-isotropic diffusion models for large datasets are not available.
>
> We appreciate the suggestions. Meanwhile, we hope the reviewer could consider that not all research groups have enough computational resources. Diffusion models are quite expensive to train and that's why we view our work as a timely contribution, but even so, some of the suggested experiments are beyond the computational capability of the authors'. We hope the reviewer could agree with us that this should not stop progress from academia.

---

### Official Review · Reviewer_4f5G · 2022-11-02

**Confidence:** 3
**Correctness:** 3
**Technical Novelty And Significance:** 3
**Empirical Novelty And Significance:** 2
**Recommendation:** 6

**Clarity, Quality, Novelty And Reproducibility:**

Overall, the paper is with good clarity (except the introduction). This paper is based on DDIM, BDM, and CLD, so has limited novelty. However, the authors showed interesting results.

**Strength And Weaknesses:**

Strengths:

1. The analysis toward understanding why DDIM accelerates the sampling process is well organized and inspiring. Following that, because complexity can be decomposed into simple cases, extending and interpreting the results in real datasets is natural. (even though in a less rigorous way) The research methodology is of merit, starting from a clear analysis of the simple and synthetic case and finally carefully verifying this on realistic datasets.
2. Given the fact that authors train an adjusted CLD, (which is not training free to achieve the best results), the final results are pretty good.

Weaknesses/Concerns:

1. To apply the acceleration techniques, one first needs to formulate irregular diffusion models into a regular form provided by this paper and make a correspondence between less structured generalized diffusion model designs with parameters and formulations in this framework, which incurs an invisible bar for many practitioners. This is somehow not an off-the-shelf acceleration method.
2. Authors spend a notable portion explaining the hyperparameters in CLD and the correspondence between the original CLD and newly designed in this paper, like the $L_t$ and $R_t$. In this case, the gDDIM is not a training free method. To achieve the best acceleration, training from scratch is necessary, which actually changes the model design for CLD.
3. The writing of the introduction can be polished much further. Without good logical guidance, it is a bit confusing for readers to see many less structured references to the method section. The authors should compress the related works in the introduction, move this part to the related works, and use the additional space here to organize and present the *logic* of this paper.

Minor comments:

1. Related works should be presented in the main paper.
2. The definition of $L_t$ should be introduced earlier in the main paper, as this is discussed and compared in the experiments but not clearly stated.
3. It should be $N$ score evaluation in Table 8, not $n$ score evaluation.

**Summary Of The Paper:**

This paper provides a generalized framework based on DDIM that explains the acceleration led by DDIM and can be extended to non-isotropic diffusion models like Blurring diffusion model (BDM) and Critically-damped Langevin diffusion model (CLD). The authors build up their insights using a toy dataset consisting of only one data point, in which the parameterization of DDIM can produce smooth network output along trajectories and thus enable large step size to achieve sampling acceleration. Then this insight is generalized to real dataset by considering the manifold hypothesis and a mixture of Dirac distributions. As an application of this framework, authors first rewrite non-isotropic BDM and CLD into gDDIMs and show the acceleration achieved by the smoothed estimation along the trajectory.

**Summary Of The Review:**

This paper is overall interesting but not appealing enough. Its merits and weaknesses are both clear. Authors make great efforts to provide a comprehensive analysis. However, there are numerous weaknesses that limit the scope of this paper. Based on the authors' response, I am happy to update my score accordingly.

---

> ### Author Response · Authors · 2022-11-15
> **Response**
>
> We appreciate the detailed reviews and thoughtful feedback from the reviewer.
>
> ********Q: This paper is based on DDIM, BDM, and CLD, so has limited novelty.********
>
> A: The main contributions of our work include an interpretation of DDIM and a generalization of DDIM (gDDIM) to general diffusion models based on this interpretation; both are novel. Our work is not based on BDM or CLD. Instead, these are examples of diffusion models we use to verify gDDIM. gDDIM can be applied to all other existing diffusion models, and can provide insights to design accelerated sampling algorithms for new diffusion models that have not been developed yet.
>
> ********Q: The method is not training-free acceleration.********
>
> A: Our method is training-free in the sense that it does not need further training once the full score model is trained. To implement gDDIM, we only need to slightly modify the parameterization of the score model, and the discretization scheme during sampling time, none of which requires further training.
>
> The reason we do not use a pre-trained CLD is that the original CLD proposed in [1] only provides **partial score instead of the full score of the noised data distribution**. The original work in CLD leverages the special structure of $\mathbf{G}_t$ (only one element in 2x2 matrix is non-zero) and only trains the partial score function in the velocity channel. In our work we find that a proper parameterization that takes advantage of the score information in both $\mathbf{v}_t$ and $\mathbf{x}_t$ channels can significantly accelerate the sampling of the diffusion model.  Unfortunately, authors of CLD do not release score models for the $\mathbf{x}_t$ channel. As a consequence, we need to train the score model from scratch in our experiments related to CLD.
>
> For other diffusion models such as DDPM and BDM with a regular $\mathbf{G}_t$, gDDIM can be applied directly to pre-trained models.
>
>
> ********Q: One first needs to formulate irregular diffusion models into a regular form provided by this paper and make a correspondence between less structured generalized diffusion model designs with parameters and formulations in this framework, which incurs an invisible bar for many practitioners. This is somehow not an off-the-shelf acceleration method.********
>
> A: The two key modifications of gDDIM include a high-order solver and a proper score parameterization $\mathbf{R}_t$ for generating samples. In the paper, we present $\mathbf{R}_t$ with exponential multistep updates in Algorithm 1.
>
> We included more details on how to apply gDDIM with pre-trained diffusion models in C.4.
>
> We included jax-based code for CLD in our supplementary materials and plan to release a generic package for gDDIM in the near future.
>
> **Q: The authors should compress the related works in the introduction, move this part to the related works, and use the additional space here to organize and present the logic of this paper. Related works should be presented in the main paper.**
>
> A: We appreciate your suggestions on improving the presentation. Due to space limitation, we include most relevant related works in the introduction and defer a more comprehensive discussion on related works in the Appendix. This practice is quite common in machine learning conferences. We have also added a paragraph at the end of the introduction to outline the organization of the paper.
>
> **Q: The definition of $\mathbf{L}_t$ should be introduced earlier in the main paper, as this is discussed and compared in the experiments but not clearly stated. And typo in Table 8**
>
> A: We thank the reviewer for pointing it out. We have made corresponding modifications to the manuscript. Since $\mathbf{L}_t$ is only used for CLD, we include it in the experiment section.
>
> [1] Tim Dockhorn, Arash Vahdat, Karsten Kreis, Score-Based Generative Modeling with Critically-Damped Langevin Diffusion

---

> > ### Comment · Reviewer_4f5G · 2022-11-28
> > **Post-rebuttal**
> >
> > I appreciate the authors' response. The response partially solved my concerns. In addition, by providing a detailed step-by-step explanation for constants and hyperparameters setup and releasing the corresponding JAX implementation afterward, the invisible bar for practitioners in vision or other domains can be reduced. I'm raising the rating to 6 accordingly.
> >
> > I am still not fully convinced about the training-free property of the current framework. As the author mentioned, this method can work when full score networks are trained and provided. If the model, like the original CLD, was designed and trained in a partial manner, it will prevent this method from achieving a reasonable acceleration, while the original sampling methods can work (but lose the acceleration, admittedly). To achieve the desired acceleration, we have to change the model design to suit this acceleration framework, which incurs the training/retraining of the original model. This retraining (not fine-tuning but training again) could be done for CLD because the model scale is not such large. But what if this model is of Stable Diffusion scale or even larger and not designed in the expected form? Should we retrain the model to fit into the requirement of acceleration design?

---

> > > ### Author Response · Authors · 2022-12-04
> > > **Response**
> > >
> > > Thank you for raising the score.
> > >
> > > To be clear, our gDDIM method is applicable **without retraining** whenever a full score network is trained and provided, **even if** the forward diffusion uses a **different parametrization** from the one we proposed. To our best knowledge, all existing works except for the original CLD work learn the full score information. The original CLD uses a very special forward noising scheme that makes it possible to generate samples without full score information. Any reparametrization of it, including the one we proposed, need to learn the full score.
> > >
> > > We would like to emphasize that it doesn’t incur much **additional computational or memory cost** to learn the full score instead of a partial score. We can use exactly the same neural network with a minor modification on the output layer, and the training cost is similar. With this in mind, we **advocate of full score training** always; it has similar cost for training but makes it more flexible for accelerating sampling. On the other hand, even in the scenario with pertained partial score models, we can finetune the score models by finetuning a few layers. This is a better option since feature extractors in pre-trained models, such as CNN kernels, do not need to be trained from scratch.
> > >
> > > We would also like to emphasize our approach is different from training-required accelerations such as distillation, where both training diffusion models and distilling models are necessary. As a comparison, gDDIM suggests a parameterization that can not only be used in denoising score models but also enjoys accelerations. In the examples we consider in the paper, gDDIM reduces to several existing training-free accelerations in DDPM, such as DDIM, DEIS. The BLD model learns the full score and thus does not need retraining to apply gDDIM. For CLD, gDDIM suggests a different parameterization, which enjoys significant accelerations with the same network architecture and a similar computational budget.

---

### Decision · Program_Chairs · 2023-01-20

**Decision:**

Accept: notable-top-25%

**Justification For Why Not Higher Score:**

Mainly the generality of the proposed accelerations is not clear. Otherwise, overall a solid paper.

**Justification For Why Not Lower Score:**

The demonstrations are quite compelling.

**Metareview: Summary, Strengths And Weaknesses:**

The work generalizes the DDIM framework and proposes acceleration ideas that are demonstrated to be effective for three examples.

At this point, it is still not clear if the training-free property holds in general cases, but the improvements in the current results suggests that the proposed ideas hold their ground in interesting subcases.

The writing of the paper is quite clear and the authors do a great job in explaining and interpreting their results.

**Note From Pc:**

if the above contains the word "oral" or "spotlight" please see: "oral" presentation means -> notable-top-5% and "spotlight" means -> notable-top-25%. As stated in our emails, we are disassociating presentation type from AC recommendations